# Lamina-specific immunohistochemical signatures in the olfactory bulb of healthy, Alzheimer's and Parkinson's disease patients

Helen C. Murray [1,2✉], Kory Johnson[3], Andrea Sedlock [4], Blake Highet[1], Birger Victor Dieriks [1], Praju Vikas Anekal[1,5], Richard L. M. Faull[1], Maurice A. Curtis[1], Alan Koretsky[2] & Dragan Maric [4✉]

Traditional neuroanatomy immunohistology studies involve low-content analyses of a few antibodies of interest, typically applied and compared across sequential tissue sections. The efficiency, consistency, and ultimate insights of these studies can be substantially improved using high-plex immunofluorescence labelling on a single tissue section to allow direct comparison of many markers. Here we present an expanded and efficient multiplexed fluorescence-based immunohistochemistry (MP-IHC) approach that improves throughput with sequential labelling of up to 10 antibodies per cycle, with no limitation on the number of cycles, and maintains versatility and accessibility by using readily available commercial reagents and standard epifluorescence microscopy imaging. We demonstrate this approach by cumulatively screening up to 100 markers on formalin-fixed paraffin-embedded sections of human olfactory bulb sourced from neurologically normal (no significant pathology), Alzheimer's (AD), and Parkinson's disease (PD) patients. This brain region is involved early in the symptomology and pathophysiology of AD and PD. We also developed a spatial pixel bin analysis approach for unsupervised analysis of the high-content anatomical information from large tissue sections. Here, we present a comprehensive immunohistological characterisation of human olfactory bulb anatomy and a summary of differentially expressed biomarkers in AD and PD using the MP-IHC labelling and spatial protein analysis pipeline.

[1] Department of Anatomy and Medical Imaging and Centre for Brain Research, Faculty of Medical and Health Science, University of Auckland, Private Bag, Auckland 92019, New Zealand. [2] Laboratory of Functional and Molecular Imaging, National Institute of Neurological Disorders and Stroke, National Institutes of Health, Bethesda, MD 20892, USA. [3] Bioinformatics Section, National Institute of Neurological Disorders and Stroke, National Institutes of Health, Bethesda, MD 20892, USA. [4] Flow and Imaging Cytometry Core Facility, National Institute of Neurological Disorders and Stroke, National Institutes of Health, Bethesda, MD 20892, USA. [5] Present address: Harvard Medical School, 77 Avenue Louis Pasteur, Boston, MA 02115, USA. ✉email: h.murray@auckland.ac.nz; maricd@ninds.nih.gov

To understand the neurobiology of human brain diseases, anatomical studies seek to label many different tissue components across large areas, including cell populations, vasculature, and disease proteins. Many proteins must be labelled on the same tissue section to capture the complexity of human brain cytoarchitecture. However, conventional immunohistochemistry (IHC) typically allows only 3–5 labels on a single tissue section due to the overlap in antibody host species and the spectral overlap of fluorophores. Multiplex labelling technology that generates spatial maps of large numbers of proteins can facilitate discoveries in neuroanatomy and the study of neurodegenerative diseases. Multiplexing increases the power and efficiency of neuroanatomical studies by maximising the amount of spatial protein data acquired from a single tissue section.

Several new multiplex immunofluorescence technologies have emerged in immunology, focussing on tumour cell phenotyping with spatial context. These techniques such as MxIF[1], OPAL[2], array tomography[3], CycIF[4] and 4i[5] are fluorescence-based protocols that use readily available antibodies and standard microscopy but involve slow iterative cycles of 1–5 antibodies and thus require weeks to capture 10–40 markers. Alternatively, DNA barcoding protocols such as CODEX[6] or SABER[7] are high throughput with 10+ antibodies labelled per cycle, but the primary antibodies must be directly conjugated to the DNA strands, which involves considerable preparation and cost. The significant investment in new reagents and imaging equipment is also a hurdle for many laboratories. Finding a compromise between the accessibility and throughput of the technique is necessary for highly multiplexed IHC to become routine in neuroanatomical studies.

To address these issues, we present a multiplexed fluorescence immunohistochemistry (MP-IHC) labelling and analysis approach for human brain tissue that improves the throughput of conventional IHC and current multiplex protocols through iterative labelling of up to 10 different antibodies per cycle, with no limitation on the number of cycles[8]. This technique uses commercially available primary and secondary antibodies, taking advantage of different host species and immunoglobulin types/subtypes with appropriately matched secondary antibodies. Each labelling round is imaged using a standard widefield epifluorescence microscope equipped with commercially available narrow bandpass excitation/emission filters to spectrally separate each fluorophore, with minimal spectral crosstalk[8]. Large tissue regions can be acquired using stitching applications within the standard microscope acquisition software. Repeated cycles of antibody stripping and relabelling are followed by alignment of images using the DAPI channel from each staining/imaging cycle for spatial registration at the pixel level. We

previously demonstrated this method on rodent brain tissue[8], and here we demonstrate the first application of this protocol for human tissue neuroanatomical studies by labelling up to 100 antibodies on formalin-fixed paraffin-embedded sections of human olfactory bulb from neurologically normal (herein referred to as 'no significant pathology' or NSP), Alzheimer's (AD), and Parkinson's disease (PD) patients. This brain region is affected in the prodromal phase of these diseases and has had limited neurochemical characterisation to date[9–12].

The human olfactory bulb has a disorganised laminar structure compared to the circumferential layer organisation observed in the rodent bulb[13]. Histological staining or chromogenic labelling is commonly used to delineate subregions within the human olfactory bulb; however, the heterogenous laminar structure and inter-individual variability in structure makes any such delineation highly subjective. In particular, delineation of the anterior olfactory nucleus (AON) is important for studies of neurodegeneration as pathological aggregates such as α-synuclein, tau, β-amyloid, huntingtin and TAR DNA-binding protein 43 (TDP43) specifically accumulate in this region of the olfactory bulb in many neurodegenerative diseases[12,14–18]. To facilitate more accurate layer delineation, we used the MP-IHC approach to generate a comprehensive neurochemical atlas of the human bulb that identifies key structures, layers and cell populations. Unlike traditional immunofluorescence, MP-IHC provides an efficient and powerful method of conducting such analyses, as the spatial context of all tissue features can be assessed on a single tissue section.

The production of such high-content anatomical information necessitates new tools for image alignment and analysis. Segmentation of image features such as individual cells or tissue layers is particularly challenging for neurological tissue due to the complex spatial arrangement of neurons, glia and their processes, and the size of human brain regions. Therefore, we developed an analysis tool that we term spatial protein analysis for unsupervised assessment of tissue features in our large stitched multiplexed images. This approach draws on current approaches for spatial transcriptomics and single-cell genomics analysis, such as the Seurat pipeline[19]. Each image is split into separate pixel bins, and for each bin, the fluorescence intensity of each antibody is determined. The measurements from each bin are combined into a 'bin count matrix' analogous to single-cell genomics analysis. An unsupervised clustering analysis can be performed to identify groups of bins with similar label signatures that correspond to distinct tissue layers or structures (Fig. 1). Differential antibody labelling can also be investigated across sections in an

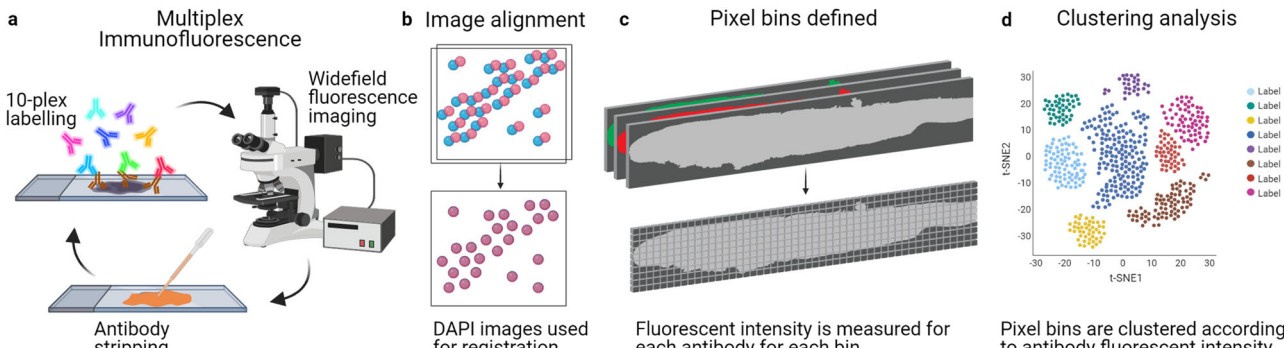

**Fig. 1 MP-IHC and spatial protein analysis methodology. a** Sections were labelled using iterative rounds of 10-plex fluorescent immunohistochemistry, imaging and antibody stripping. **b** The images from each labelling round were manually aligned in photoshop using the DAPI channel from each round as a reference. **c** The aligned images were imported into R, and 10 × 10 μm pixel bins were calculated with the x-y coordinates of each bin captured. The fluorescence intensity of each antibody label was measured for each pixel bin. **d** Spatial protein analysis was performed using R and the Seurat library to generate UMAP plots of bin clusters and identify differential markers between disease groups. Created with BioRender.com.

unsupervised manner. We demonstrate the application of this spatial protein analysis pipeline to define anatomical features of the human olfactory bulb and identify differences between disease and non-disease groups with minimal user input. Together, our MP-IHC and spatial protein analysis pipelines offer a powerful and easily customisable platform for neuroanatomical discovery on human tissue that improves throughput and maintains accessibility for routine use.

## Results

### Efficient neurochemical characterisation of the human olfactory bulb using multiplex IHC.
Formalin-fixed paraffin-embedded sections of human olfactory bulb were processed using 10 iterative rounds of 8-10-plex IHC staining to label markers that map the olfactory bulb cytoarchitecture (Supplementary Fig. 1, antibodies detailed in Supplementary Table 1). Across the labelling rounds, we also tested antibodies from different manufacturers directed to the same antigen, and some antibodies were repeated across different labelling rounds. We selected 28 different markers that showed robust labelling across all sections for image analysis (Supplementary Fig. 2). The aligned images reveal the human olfactory bulb complex laminar organisation, which is not readily observed using traditional histology or single chromogenic labelling (Fig. 2a, b). Our qualitative assessment of the aligned images noted several important features of the human bulb cytoarchitecture and laminar structure. We observe that the olfactory bulb layers contain distinct neuronal populations based on their neurochemical phenotype (Fig. 2c). Tyrosine hydroxylase+ neurons are confined to the periglomerular space, while calbindin labels tufted cells of the external plexiform layer and neurons within the AON. Calretinin labelled small periglomerular cells and glomeruli as well as a subset of granule cells (Fig. 2h). The neuronal phenotyping also revealed at least three neurochemically distinct granule cell populations in the human olfactory bulb: neurogranin+, calretinin+ and neurogranin-/calretinin- cells (Supplementary Fig. 3).

The MP-IHC labelling also enabled a more accurate delineation of the olfactory bulb layers. We noted that the AON consists of multiple compartments that can be reliably delineated using several markers, the most effective being protein gene product 9.5 (PGP9.5) and the relative absence of glial fibrillary acidic protein (GFAP) and S100 (Fig. 2b, d, g). The lateral olfactory tract, which borders the AON, is a dense fibre tract with high expression of S100, GFAP, myelin basic protein and 2′,3′-Cyclic-nucleotide 3′-phosphodiesterase (CNPase; Fig. 2b, d, e). The granule cell layer (GCL) is most clearly delineated by the high density of DAPI nuclei and randomly orientated myelinated axons (Fig. 2c, e). The external plexiform layer could be divided into two sub-layers based on myelin basic protein and CNPase labelling, where the deeper sub-layer (d-EPL) bordering the mitral cell layer is relatively devoid of myelinated axons and the more superficial sub-layer (s-EPL) bordering the glomerular layer consists of randomly orientated myelinated axons (Fig. 2e). In the lateral olfactory tract, blood vessels were predominantly longitudinal to the sagittal plane, while there was a mixture of longitudinal and transverse vessels in the GCL and EPL (Fig. 2f). Lastly, the glomeruli are specifically labelled by olfactory marker protein (OMP), while Ulex Europaeus Agglutinin I (UEA-I) Lectin, microtubule-associated protein 2 (MAP2), PGP9.5, tomato lectin, neural cell adhesion molecule (NCAM) and synaptophysin all label the glomeruli as well as other tissue structures (Fig. 2g, Supplementary Fig. 2). By using many markers, MP-IHC can be used to investigate the composition of specific structures such as myelinated axons (Fig. 2j), the blood-brain barrier (Fig. 2k) and the periglomerular region (Fig. 2l). A summary of marker signatures for each olfactory bulb layer are summarised in Table 1.

Qualitative comparison of NSP, AD, and PD olfactory bulbs labelled using MP-IHC showed a thinner white matter tract in AD and PD compared to NSP cases. Glomeruli also appear to be smaller and less dense in the PD cases, and pathological aggregates of tau and α-synuclein were concentrated within the AON of AD and PD cases, respectively (Supplementary Fig. 4).

### Spatial protein analysis provides unsupervised detection of tissue layers and components.
Accurate and consistent region delineation is essential in anatomical studies to ensure that cell number or subregion volume comparisons between individuals and disease groups are meaningful and reliable. Therefore, we developed an approach using MP-IHC to provide reliable delineation of human olfactory bulb structures and layers that would enable accurate, automated detection of tissue layers and structures.

To facilitate unsupervised detection of tissue features, the images from each antibody label were split into bins of $31 \times 31$ pixels ($10 \, \mu m^2$) and the median intensity value (between 0 and 1) per marker was determined for each bin and multiplied by 100 to produce a bin matrix per bulb. The Seurat R package was used to perform dimension reduction and clustering of the bins from each section. Plotting the x-y coordinates for individual bins within each cluster revealed distinct spatial locations that correspond to specific tissue layers or features such as glomeruli and blood vessels (Fig. 3a, c, Supplementary Fig. 5). A heat map of the Z-score of labelling for each antibody per cluster was used to interpret the antibody composition of each cluster (Fig. 3b), and qualitative examination of the spatial distribution of bins within the cluster compared to the original MP-IHC images was used to determine the layers (Supplementary Fig. 6).

We determined the most conserved markers for each bulb layer by assessing each cluster's antibody composition and spatial distribution (Table 1, Supplementary Fig. 6). For each tissue section, we then selected the cluster that had the highest expression of the signature markers for each layer and qualitatively compared the spatial distribution of that cluster to the original MP-IHC labelling (Fig. 4). This comparison confirmed that the marker signature for each bulb layer is consistent between sections and between disease groups.

### Spatial protein analysis highlights differential abundance of antibody labelling between NSP, AD and PD groups.
To demonstrate the use of spatial protein analysis to investigate differentially abundant markers, we compared the number of bins labelled for each antibody between groups. As we analysed one mid-sagittal section of the olfactory bulb per case, the spatial protein analysis results should be considered an indicative screen for differences in marker abundance between AD, PD, and NSP cases, with further validation required using additional tissue sections. To account for differences in absolute fluorescence intensity between sections, the bins within each section were thresholded for each label based on Poisson distribution Z-score > 0.05 across that section (Supplementary Fig. 7). The ratio of positive bins: total bins was determined for each antibody per section and compared between groups. Of the markers tested, calretinin, synaptophysin and tyrosine hydroxylase were differentially expressed between groups at $P < 0.1$, while beta-amyloid, GFAP, MAP2, S100 and tau were differentially expressed at $P < 0.05$ (Fig. 5a, Supplementary Data 1, Supplementary Fig. 8).

The cases were then clustered by principal component analysis of the positive bins: total bins ratio for each label. The cases clustered into three clear groups with 4 of 5 PD cases clustering, 3 of 4 NSP cases clustering and all 5 AD cases clustering together (Fig. 5b). One PD and one NSP case clustered with the AD cases. Interestingly, pathological assessment of this PD case (PD58)

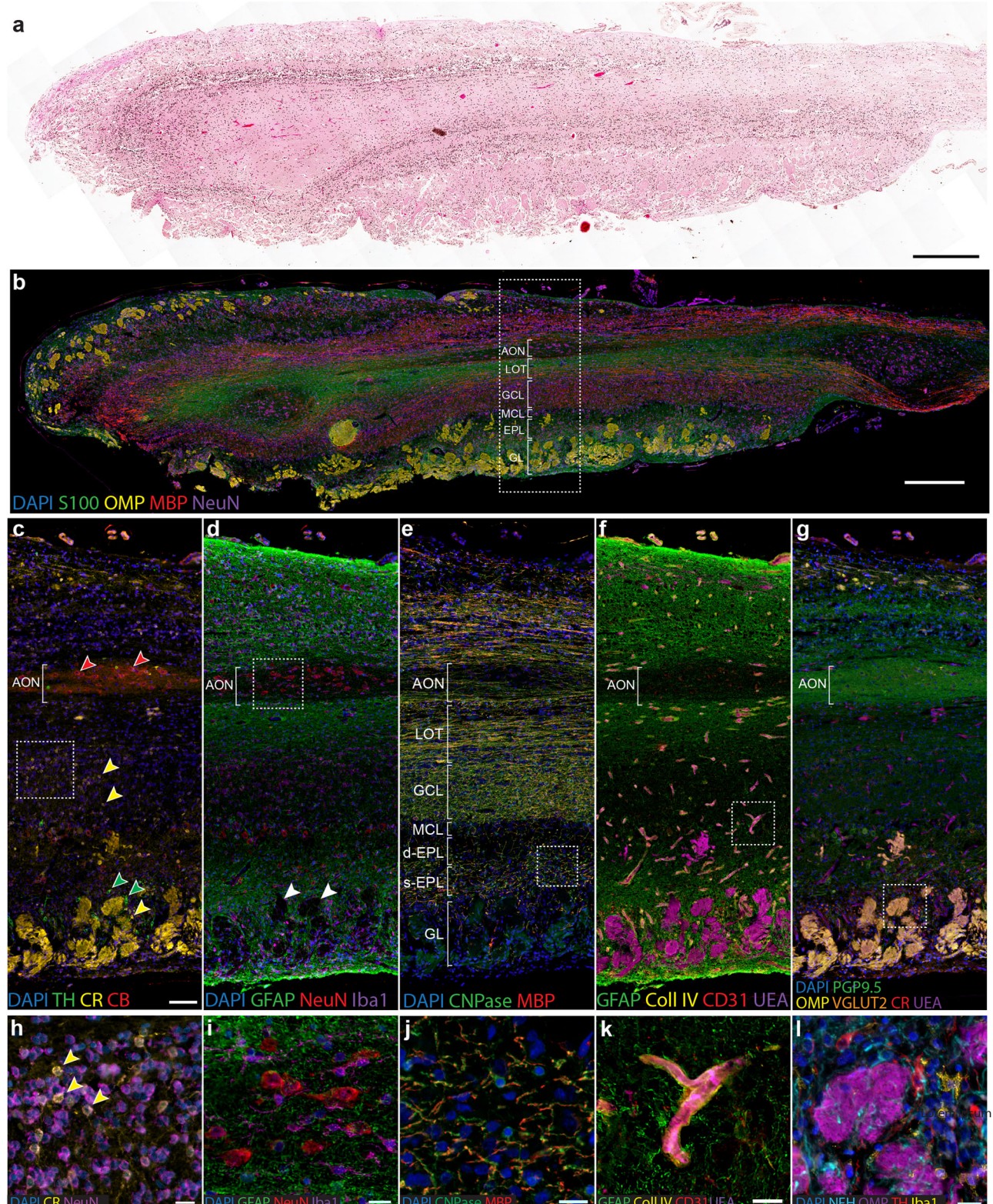

identified intermediate AD pathology severity as well as neocortical Lewy body disease.

Post-hoc testing showed tau and β-amyloid were increased in AD compared to NSP and PD cases, while GFAP was increased in PD and AD compared to NSP cases. MAP2 was increased in PD compared to NSP and AD cases and S100 was decreased in AD compared to NSP cases. In line with previous studies, tyrosine hydroxylase was increased in PD[17] and calretinin was decreased in AD and PD cases[20] relative to NSP cases (Fig. 5c).

**Sub-clustering of tau-positive bins identifies markers associated with tau pathology in AD cases.** To further investigate the spatial and neurochemical signature of AD cases, we determined

**Fig. 2 Overview of MP-IHC labelling of the human olfactory bulb.** NSP human olfactory bulb stained with H&E (**a**) and MP-IHC (**b**) illustrating that the laminar structure is easily identified using a combination of markers. Magnified region marked by the dotted box in (**b**) illustrating the laminar organisation of different cell populations (**c**, **d**, **g**), axon tracts (**e**) and blood vessels (**f**). **c** Tyrosine hydroxylase (TH), calretinin (CR), and calbindin (CB) labelling identify different neuronal populations within layers of the olfactory bulb. Tyrosine hydroxylase+ neurons (green arrows) are periglomerular, calretinin (yellow arrows) identifies a subpopulation of granule cells, periglomerular cells and the glomeruli, and calbindin (red arrows) labels AON neurons. **d** Distribution of astrocytes (GFAP), microglia (Iba1) and neurons (NeuN), illustrating the relative lack of astrocyte processes in the glomeruli (white arrows) and AON, and ubiquitous distribution of microglia. **e** Orientation of myelinated axons in each layer. Within the external plexiform layer (EPL) and granule cell layer (GCL), myelinated fibres are randomly orientated. The deep external plexiform (d-EPL) sublayer can be delineated based on the relative lack of myelinated axons compared to the superficial (s-EPL) sublayer. The lateral olfactory tract (LOT) surrounding the AON consists of a longitudinal axon tract travelling toward the cortex. **f** Blood vessels are longitudinal to the sagittal plane in the LOT and transverse in the GCL and EPL. **g** Glomeruli labelled using a range of antibodies including PGP9.5, OMP, VGLUT2, calretinin and UEA-I lectin. PGP9.5 also allows for delineation of the AON boundary. (**h**) Magnified region marked by the dotted box in (**c**) illustrating the calretinin+ subpopulation of granule cells. (**i**) Magnified region marked by the dotted box in (**d**) showing distribution of astrocytes (GFAP), microglia (Iba1) and neurons (NeuN) in the AON. (**j**) Magnified region marked by the dotted box in (**e**), CNPase+ oligodendrocyte processes wrapped around myelinated axons illustrating the structure of these axons. **k** Magnified region marked by the dotted box in (**f**), MP-IHC allows for simultaneous visualisation of the blood-brain barrier components, including collagen IV+ basement membrane, UEA-I lectin+ endothelium, CD31+ endothelial cells and GFAP+ astrocytes. **l** Magnified region marked by the dotted box in (**g**), the glomerular microenvironment: neurofilaments within OMP + glomeruli, surrounded by tyrosine hydroxylase+ periglomerular cells and Iba1+ microglia. Scale bars (**a**, **b**) 500 µm, (**c–g**) 100 µm, (**h–l**) 20 µm.

**Table 1 Marker signatures for olfactory bulb layers.**

| Layer | Markers |
|---|---|
| Anterior olfactory nucleus | PGP9.5, calbindin, synaptophysin |
| Lateral olfactory tract | GFAP, S100, neurofilament light, neurofilament heavy, PGP9.5, myelin basic protein |
| Granule cell layer | DAPI, histones |
| External plexiform layer | GFAP, synaptophysin |
| Glomerular layer | OMP, UEA-I lectin, PGP9.5, MAP2 |

the combination of labels that co-occurred with tau. Each bin in the count matrix was assigned a binary string signature which indicated whether each marker is present or absent in the bin after the Poisson distribution Z-score > 0.05 thresholding for that section. The number of bins with each string signature were tallied for each section and for strings that were differentially expressed between groups, the bins were plotted according to their x-y coordinates (Fig. 6a). These slide plots show that the differentially expressed tau+ bins were predominantly located in the same anatomical region in AD. The label signatures of these differentially expressed strings were permutations of the same eight markers, PGP9.5, MAP2, neurofilament light, synaptophysin, calbindin, NeuN, histones and GFAP. These markers predominantly labelled the AON, indicating that differentially expressed tau bins are concentrated in the AON in AD (Fig. 6b). Overall, this approach demonstrates the potential for spatial protein analysis to identify combinations of co-labelled markers in MP-IHC labelled tissue, determine whether their abundance differs between groups and map them back to their anatomical space.

## Discussion

MP-IHC labelling is an efficient and powerful technique to study the heterogeneity and complexity of brain structure in aging and disease. Our MP-IHC approach improves the throughput of current methods but maintains accessibility in set-up time and cost by using commercially available reagents, standard immunofluorescence labelling protocols and standard widefield epifluorescence microscopy with relatively low-cost modifications.

This MP-IHC approach uses iterative rounds of 10-plex labelling and can achieve 100 (or more) antibodies on a single tissue section, enhancing the potential to continue increasing the dimensionality of the spatial protein analysis. Here we have used as many as 81 antibodies or labels on a single tissue section, some of which were duplicates or different antibodies to the same antigen allowing additional validation for the expression of selected targets. Twenty-eight antibodies were selected for quantitative analysis. Such high-dimension anatomical data can be used for in-depth studies of specific protein relationships with a spatial context or more broad screening studies. In this study, we demonstrated the versatility of MP-IHC using a range of antibody panels for cell phenotyping together with antibodies that identify olfactory bulb-specific structures. Due to the abundance of commercially available antibodies raised in various species, antibody panels can be easily customised according to the brain region or the biological question. Furthermore, as the MP-IHC approach is carried out on formalin-fixed paraffin-embedded tissue, it is compatible with banked tissue and tissue prepared for pathology assessment. This versatility makes the approach accessible to a wide range of researchers for many different applications.

Using the MP-IHC approach, we have presented a comprehensive overview of human olfactory bulb cytoarchitecture and neurochemical anatomy by labelling a wide range of markers on a single tissue section. This labelling agrees with previous reports of human olfactory bulb neurochemical anatomy and provides new insights into human bulb anatomy. We observed neuronal markers such as calbindin, PGP9.5, tyrosine hydroxylase and calretinin demonstrate the distinct laminar locations of different neuronal populations within the olfactory bulb. This agrees with previous studies that have described the laminar distribution of these neuronal markers using single immunoperoxidase labelling or triple immunofluorescence[17,20–28]. We observed calbindin labelled large neurons in the mitral cell layer and AON, which agrees with previous reports[18,21]. The distribution of glial markers such as Iba1 and GFAP has also been demonstrated previously[29–31], as have a range of markers that label glomeruli, including OMP, VGLUT2, NCAM, PGP9.5, synaptophysin and UEA-I lectin[11–13,24,27,28,32]. Thus, the distributions of the markers that we labelled using MP-IHC agrees with previous literature, with the added advantage that the relative distributions of all these markers can be observed on the same tissue section. One new insight we observed is the distinct granule cell populations within the human olfactory bulb that label with different permutations of calretinin, NeuN and neurogranin. This aligns with previous studies of granule cell diversity in the rodent olfactory bulb that show distinct populations of calretinin+, neurogranin+ and 5T4+ granule cells, which are speculated to mediate experience-dependent plasticity as well as olfactory detection and discrimination[33–35].

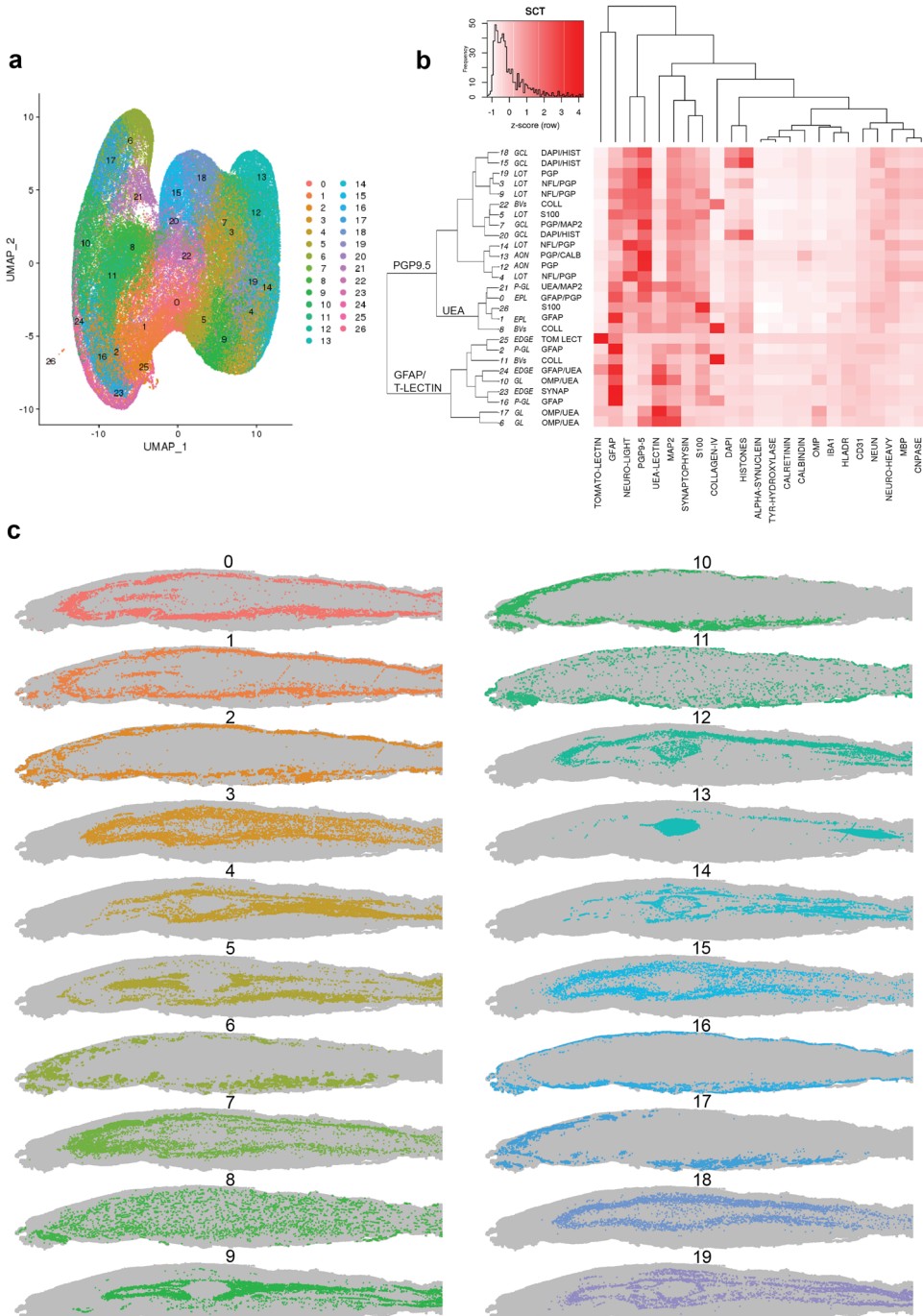

**Fig. 3 Individual section cluster analysis for NSP case H190. a** UMAP of pixel bin data coloured by cluster. **b** Heat map of z-score of labelling intensity for markers within each cluster. Clusters are organised based on a hierarchical tree. Each cluster is identified by a number and a three-letter code indicating the layer it is localised to, based on visual inspection of the slide plot for that cluster, as well as the one or two most intensely labelled or distinct markers for that cluster. **c** Individual slide plots of clusters 0–19 illustrating the unique spatial distribution of each cluster. AON anterior olfactory nucleus, BVs blood vessels. EPL external plexiform layer, GL glomerular layer, GCL granule cell layer, LOT lateral olfactory tract, P-GL peri-glomerular.

The MP-IHC labelling also revealed more detail of the human olfactory bulb laminar structure than can be observed with traditional histology or IHC. For example, the deep and superficial subdivisions of the EPL were clearly delineated by CNPase and myelin basic protein. These subdivisions have been described in the rodent olfactory bulb, with the deep EPL containing the soma of tufted cells and the superficial EPL containing interneurons and their network of processes[36]. We also clearly identified the lateral olfactory tract as the region bordering the AON with high

GFAP and S100 immunoreactivity and a dense myelinated axon tract. Based on rodent literature, this region consists of mitral and tufted cell afferent fibres and is not often described in human olfactory bulb studies as it is difficult to delineate with single markers. Our MP-IHC labelling also shows the layer-specific orientation of neurofilaments and blood vessels throughout the bulb.

For this high-content anatomical data, we developed a spatial protein analysis that provides a versatile and unsupervised analysis

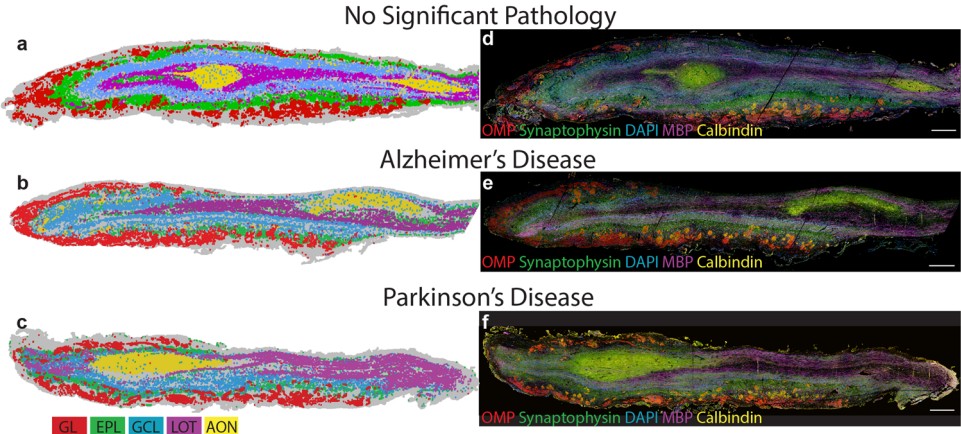

**Fig. 4 Olfactory bulb layer signatures determined by individual section cluster analysis are consistent between cases.** For each section, clusters were assigned to one of five layers (glomerular layer, GL; external plexiform layer, EPL; granule cell layer, GCL; lateral olfactory tract, LOT; or anterior olfactory nucleus, AON) based on visual inspection of the slide plot. The markers with the highest z-score of labelling intensity were noted for all clusters within a layer and the most conserved markers for each layer were determined to be the layer signature. The clusters with highest expression of signature markers for each layer are overlaid for one representative NSP (**a**), AD (**b**) and PD (**c**) case. The clusters have been pseudo-coloured by layer. The immunolabelling for one signature marker from each layer is also presented (**d–f**). The layer signature is consistent between cases and disease groups. Scale bars 500 μm (**a–f**).

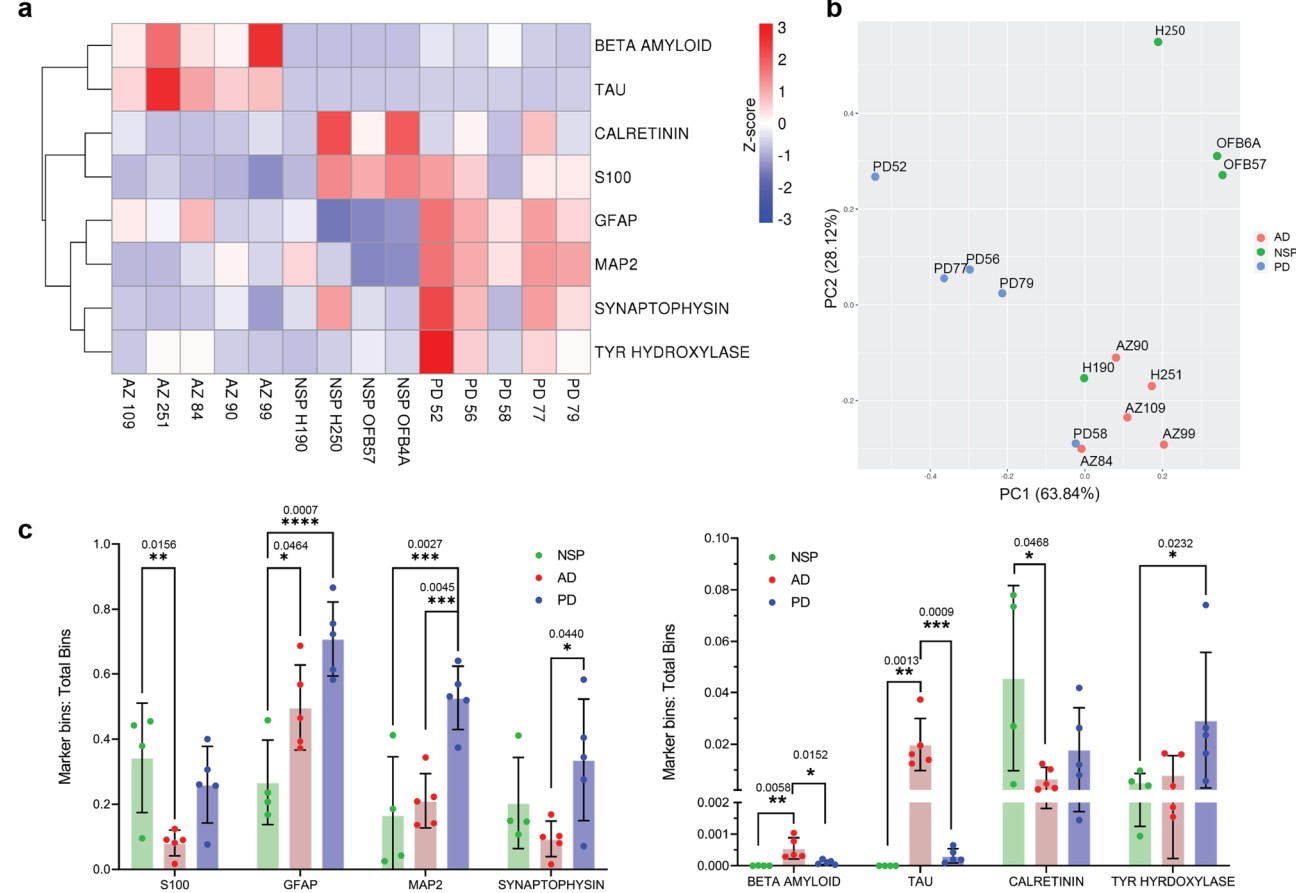

**Fig. 5 Analysis of differentially expressed markers between NSP, AD and PD bulbs.** The ratio of labelled bins to total bins was determined for each marker and each case. Markers were determined to be differentially expressed based on an uncorrected ANOVA *P* value < 0.1. **a** Heat map of the ratio of labelled bins to total bins for differentially expressed markers for each case. **b** PCA scatterplot of cases coloured by group identity. **c** Graph of labelled bins to total bins ratio per case for differentially expressed markers, n = 4 NSP cases, 5 AD cases and 5 PD cases. Error bars are standard deviation of ratio for each case within the group. *$P \leq 0.05$, **$P \leq 0.01$, ***$P \leq 0.001$, ****$P \leq 0.0001$. Exact *P* values are stated above the comparison.

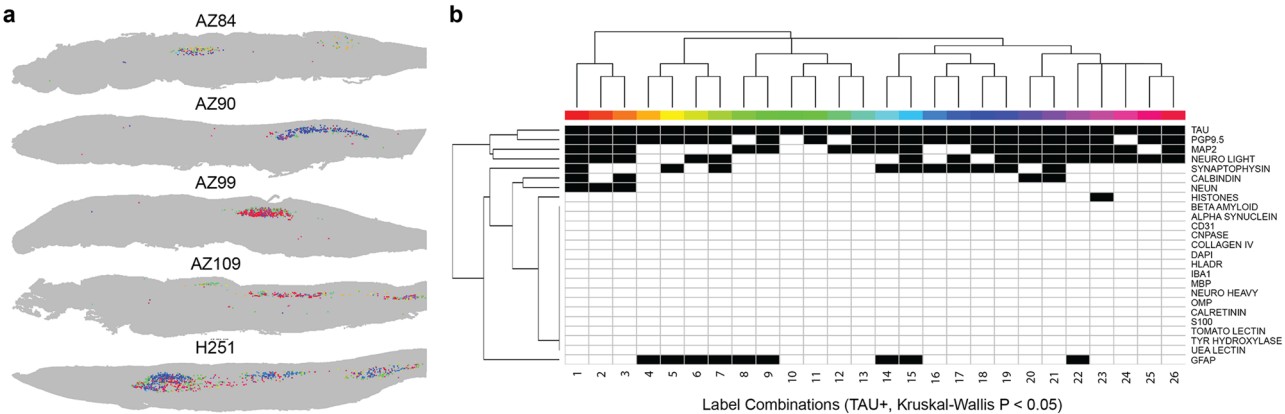

**Fig. 6 Tau string analysis for AD cases.** The combinations of markers that co-occur with tau were investigated by generating a binary string marker signature. To do this, a Poisson $P$ value was determined for each marker and each bin in the 'pseudo' count matrix and the bins with a $P$ value < 0.05 were coded as '1' while the rest were '0'. For each bin, the binary values for each marker were concatenated into a string and the frequency of each string was tested for differences between bulbs using a Kruskal-Wallis test. Strings with a $p$ value < 0.05 were determined to be differential between bulbs and plotted here. **a** Slide plots of differentially expressed tau+ pixel bins plotted according to their x-y coordinates for each AD case, coloured by tau string identity. These tau+ bins are in discrete areas of the bulb that correspond to the AON. **b** Heat map indicating the marker combinations that are present in the differentially expressed strings containing tau. Various permutations of the same eight markers are determined to be differentially expressed.

method for large image data sets. Using a pixel-bin approach rather than a cell segmentation approach allowed us to assess various tissue structures such as blood vessels, white matter, glomeruli and the olfactory bulb layers using markers that do not have a nuclear or cytoplasmic distribution. Therefore, the spatial protein analysis is a useful tool to screen a variety of markers within large multiplex image data sets. Depending on the biological question, the pixel bin size can be adjusted to modify the resolution of the approach. In this study, we selected a 10 μm² bin to achieve single cell-level resolution as some layers of the bulb (mitral cell layer and internal plexiform layer) are particularly thin. This approach improves the resolution of current spatial transcriptomics approaches such as Visium (10x Genomics), which uses 55 μm-diameter array spots[37]. Our unsupervised clustering of pixel bins based on marker intensity provides a novel approach to anatomical delineation. Here we show a distinct neurochemical signature for each bulb layer, which allows for reliable and consistent delineation necessary to study neurodegeneration. Delineation of the AON using PGP9.5, synaptophysin, and calbindin is particularly relevant as pathological aggregates accumulate in this region during the prodromal phase of AD and PD. Tau, beta-amyloid and α-synuclein aggregates are hypothesised to spread from the AON to downstream brain regions as the disease progresses[38]. Previous studies delineated the AON as discrete compartments comprising clusters of large neurons using traditional histological labelling[30,38–42]. Individual observers draw different AON boundaries based on these neuron clusters; however, our MP-IHC labelling shows that the compartments include a boundary of PGP9.5+ and synaptophysin+ neuropil that surrounds the neuron clusters, borders the axon tracts and potentially links the AON compartments together. If a neuropil region indeed connects the AON compartments, it would provide a direct route by which disease aggregates could diffuse from the olfactory bulb to the rest of the brain. Furthermore, AON delineation will permit more accurate studies of AON neuronal loss in these diseases. Estimates of AON neuron density are limited by the considerable inter-individual variability in AON shape and size and the inconsistent delineation of AON boundaries by different observers. Using MP-IHC for delineation will therefore improve the reproducibility of these studies.

Our MP-IHC and spatial protein analysis pipeline can also be used to screen whole tissue sections for differences in individual markers and pixel bins with specific marker combinations between disease groups. Here we have demonstrated this application on a single mid-sagittal section from a small sample of AD, PD and NSP cases. However, it should be noted that the bulb is a three-dimensional structure and the differentially expressed markers we present require additional validation with more in-depth neuroanatomical studies using higher numbers of cases and additional tissue sections per case. The differentially expressed markers we detected using the spatial protein analysis agree with previous human olfactory bulb studies. Our analysis showed an increase in tau in the AD olfactory bulbs, and GFAP in the AD and PD olfactory bulbs, which is indicative of tau neuropathology and astrogliosis as previously described[11,30,43]. Calretinin was reduced in both disease groups which agrees with a previous report in the PD olfactory bulb[20]. Loss of calretinin interneurons has not been investigated in human AD olfactory bulbs but are reduced in the olfactory bulb of transgenic AD mouse models, suggesting these cells may be more vulnerable to degeneration[44,45]. Similarly, tyrosine hydroxylase was increased in the PD olfactory bulb, which also agrees with previous literature[17,46]. As the dopamine and gamma aminobutyric acid (GABA) released by periglomerular tyrosine hydroxylase interneurons have an inhibitory effect on transmission between the olfactory sensory neurons and the mitral cells within the glomeruli, the increase in tyrosine hydroxylase cells is hypothesised to contribute to the olfactory impairment observed in PD[17,46]. These earlier studies used single-marker IHC and manual cell counting, requiring many tissue sections and considerable time to investigate potential differentially expressed markers. In contrast, our MP-IHC and spatial protein analysis approach provide an efficient pipeline to screen for changes on a single tissue section.

With the spatial protein analysis, we also demonstrated how principal component analysis of differentially expressed markers could be used to cluster cases by disease group. Interestingly, one AD and one PD case used in this study showed both tau and alpha-synuclein pathology in the AON and clustered together in this analysis. The AD case AZ84 had a clinical and pathological diagnosis of Alzheimer's disease with comorbid neocortical Lewy body disease and the PD case PD58 had a clinical diagnosis of Parkinson's disease with a pathological diagnosis of neocortical Lewy body disease and intermediate Alzheimer's pathology. This highlights the complexity of studying human brain tissue and neurodegenerative disease and demonstrates how the MP-IHC and spatial protein analysis could be used to group cases based on

anatomical features. Comorbid proteinopathy has been previously described in the olfactory bulb, particularly in AD. In a study of 107 AD cases, all with olfactory bulb tau pathology, Josephs et al., (2016) observed TDP43 in 15% and alpha-synuclein in 35% of the AD cases[47]. Furthermore, comorbid proteinopathy is common in the olfactory bulb of cases with multiple aggregate pathologies in the rest of the brain. Fujishiro et al., (2008) observed 93% of cases (38/41) classified as AD with amygdala Lewy bodies had alpha-synuclein in the olfactory bulb and that tau and alpha-synuclein co-localised within olfactory bulb neurons and neurites[48]. Similarly, Beach et al., (2009) observed alpha-synuclein pathology in the olfactory bulb of 88% (37/42) of AD cases with Lewy bodies and in 18% (19/103) of AD cases without Lewy bodies[49]. Description of tau and alpha-synuclein co-occurrence in PD is limited. Tau has been described in the olfactory bulb of 4/5 PD cases by Carmona-Abellan et al., (2021) and in all 6 PD cases studied by Mundiñano et al., (2011), which had tau Braak stages between III and IV[17,29]. Despite the prevalence of comorbid protein pathology in AD and PD, we still have little understanding of how these different aggregated proteins may interact and contribute to disease pathogenesis. MP-IHC offers an efficient approach to screen tissue for a range of pathological protein aggregates and will be a useful tool for investigating the role of comorbid pathologies in neurodegenerative diseases.

As tau pathology is well studied in the AD olfactory bulb, we used the spatial protein analysis approach to investigate the marker combinations that co-labelled with tau and were significantly different between NSP and AD cases. Qualitative observation of tau in the AD and PD sections indicated that the aggregates predominantly affect the large calbindin+/NeuN+ neurons within the AON. The spatial protein analysis of tau co-labelling showed that differentially expressed tau+ bins co-labelled with permutations of markers that were predominantly distributed within the AON, including PGP9.5, calbindin, MAP2 and synaptophysin. Mapping these bins back to their original coordinates revealed the AON as the primary site of differentially expressed tau in all AD cases, which agrees with the qualitative observation. As such, we demonstrated how this analysis is a versatile and scalable approach that could be applied to other differentially expressed markers to screen for interesting co-labelling combinations in disease tissue.

Additional considerations are required for MP-IHC labelling in neuroanatomical studies. Delicate tissue may be susceptible to damage caused by the iterative processes of coverslip removal, antibody stripping, and antigen retrieval. Therefore, great care must be taken during these steps to preserve the tissue. Some antigens may also be degraded in the later rounds of labelling, leading to variable fluorescence intensity. Our incremental antigen retrieval conditions using iterative heat-mediated antigen retrieval steps in Tris-EDTA pH9 buffer were designed to progressively unmask target antigens in the same tissue samples, whereby the most heat-labile tissue targets could be probed in early rounds of MP-IHC, after minimal antigen retrieval steps, and more robust targets are probed in later rounds after more extensive antigen retrieval steps. It is therefore important to validate that the antibodies used produce the same labelling distribution in both single and multiple round labelling experiments. All antibodies used for this study have been previously validated for use in formalin-fixed paraffin-embedded rodent and human brain tissues as cited in our previous publications[8,11,12,18,27,50,51] and our MP-IHC labelling was consistent with that observed using single round labelling in these previous studies. For quantitative comparisons, tissue sections need to undergo the same labelling protocol and only labelling from the same round should be compared across different tissue sections.

Future development of this spatial protein analysis will involve single-cell segmentation for clustering analysis. This type of

image segmentation would permit automated and unsupervised quantification of cell populations. Single-cell segmentation has broad applicability to many biological questions, such as more targeted interrogation of the populations most affected by pathological protein aggregates. Accurate segmentation of single cells in human brain tissue is challenging due to the variable labelling intensity of different nuclei and the inability to delineate cell boundaries due to the complicated network of cell processes labelled by membrane markers. We previously demonstrated a single nuclei segmentation pipeline that uses multiple nuclear markers for more accurate segmentation[8]. With MP-IHC, these additional markers can be added to the antibody panels to enable more accurate nuclei segmentation. This scalable approach will permit cell phenotyping analysis of MP-IHC labelled human brain tissue.

In summary, we demonstrate an efficient, versatile, robust and accessible MP-IHC approach on human brain tissue from neurological normal aged cases and disease cases. We also present a spatial protein analysis approach to screen tissue for anatomical features and disease changes in an unsupervised manner that preserves spatial context. This pipeline has significant potential to increase the power of neuroanatomical studies and can be easily adopted due to the use of conventional microscopes and reagents. Image visualisation and analysis can be performed with readily available software that does not have image size limitations (Adobe Photoshop and R). We expect this MP-IHC approach to be helpful to a wide range of researchers and applicable to various biological questions, from broad studies of regional cell phenotyping or tissue atlases to specific interrogations of disease pathology.

## Methods

**Human tissue acquisition and processing**. The post-mortem human olfactory bulbs were provided by the Neurological Foundation Human Brain Bank and the Human Anatomy Laboratory within the Department of Anatomy and Medical Imaging at the University of Auckland, New Zealand. The tissue was donated with informed consent from the family before brain removal, and all procedures were approved by the University of Auckland Human Participants Ethics Committee (Ref: 011654). The AD, PD or NSP diagnosis was determined by an independent pathologist who carried out a neuropathology assessment of the brains for each case (Table 2). The four neurologically normal (NSP) cases had no history of neurological abnormalities, and no other neuropathology was noted. The five AD cases had a clinical history of dementia, and the clinical AD diagnosis was confirmed by independent pathological assessment. The presence of tau pathology in the olfactory bulb was previously verified[11]. The five PD cases had a clinical diagnosis of PD, which was confirmed by independent pathological assessment. The presence of alpha-synuclein pathology in the olfactory bulb was previously verified[12]. The olfactory bulbs were fixed in 15% formaldehyde in 0.1 M phosphate buffer for 24 h at room temperature and paraffin-embedded in a sagittal orientation as previously described[27]. One 10 μm-thick, mid-sagittal section per case was used for this study.

**Multiplex immunohistochemistry**. Multiplex fluorescence immunohistochemistry (MP-IHC) was performed as previously described[8]. 10-plex labelling was achieved by combining antibodies from each available IgG subclass of mouse antibody (IgG1, IgG2a, IgG2b, IgG3), plus one mouse IgM antibody, plus one IgG (or IgY) antibody raised from each of the following non-mouse hosts (rat, hamster, rabbit, guinea pig, chicken, sheep, directly conjugated goat). The MP-IHC was carried out on 10 μm-thick paraffin sections of olfactory bulb tissue samples using antibody panels targeting relevant biomarkers to map the olfactory cytoarchitecture, as listed in Supplementary Table 1. The 10-plex antibody panels were designed to be internally controlled for, whereby each antibody used in a given staining round predominantly targeted a spatially distinct sub-cellular location or tissue structure, with each antibody visualized using a different spectrally non-overlapping fluorophore, allowing for empirical assessment of antibody cross-reactivity and the effectiveness of antibody removal after antibody stripping step in between each staining round, as demonstrated in Supplementary Fig. 1. Briefly, sections were first deparaffinised and treated using a standard antigen unmasking step in 10 mM Tris/EDTA buffer pH 9.0. Sections were then blocked with Human BD Fc Blocking solution (BD Biosciences) and treated with True Black Reagent (Biotium) to quench intrinsic tissue autofluorescence. The sections were then immunoreacted for 1 h at RT using 1 μg/ml cocktail mixture of immunocompatible antibody panels (see Supplementary Table 1 for antibody sources and technical specifications). This

## Table 2 Summary of cases used for this study.

| Case Number | Diagnosis | Age (years) | Sex | PMD (hrs) | NIA-AA score (AD severity) | Tau Braak Stage | B-amyloid Thal Stage | Alpha-synuclein pathology |
|---|---|---|---|---|---|---|---|---|
| AZ84 | AD | 82 | M | 18.5 | A3 B2 C1 (intermediate) | III-IV | 5 | Neocortical LBD |
| AZ90 | AD | 73 | M | 4 | A3 B3 C2 (high) | V | 5 | Amygdala-predominant LBD |
| AZ99 | AD | 94 | F | 8.5 | A3 B3 C2 (high) | V-VI | 5 | Neocortical LBD |
| AZ109 | AD | 90 | F | 31 | A3 B2 C1 (intermediate) | III | 4 | No α-synuclein |
| H251 | AD | 77 | M | 11.5 | A2 B2 C3 (intermediate) | III | 5 | No α-synuclein |
| PD52a | PD | 84 | M | 5 | - | No tau | NA | Neocortical LBD |
| PD56 | PD | 74 | M | 10.5 | - | +c | No β-amyloid | Neocortical LBD |
| PD58 | PD | 82 | F | 18 | A3 B2 C2 (intermediate) | III | 5 | Neocortical LBD |
| PD77 | PD | 76 | F | 6.5 | - | No tau | No β-amyloid | Limbic LBD |
| PD79 | PD | 77 | M | 6.5 | - | No tau | 5 | Neocortical LBD |
| H190a | NSP | 72 | F | 19 | - | No tau | NA | No α-synuclein |
| H250 | NSP | 93 | M | 19 | - | I-II | 2 | No α-synuclein |
| OFB6Ab | NSP | 82 | M | 24 | NA | NA | NA | NA |
| OFB57b | NSP | 63 | M | 36 | NA | NA | NA | NA |

aNo β-amyloid Thal stage provided.
bOnly olfactory bulbs were obtained for these cases. No pathology scores available.
cTau neuropil threads seen in CA1, but not entorhinal or middle temporal gyrus.
AD Alzheimer's disease, PMD post-mortem delay, LBD lewy body disease, NA not available, NSP no significant pathology, PD Parkinson's disease.

step was followed by washing off excess primary antibodies in PBS supplemented with 1 mg/ml bovine serum albumin (BSA) and staining the sections using a 1 µg/ml cocktail mixture of the appropriately cross-adsorbed secondary antibodies (purchased from either Thermo Fisher, Jackson ImmunoResearch or Li-Cor Biosciences) conjugated to one of the following spectrally compatible fluorophores: DyLight 405, Alexa Fluor 430, Alexa Fluor 488, Alexa Fluor 546, Alexa Fluor 594, Alexa Fluor 647, PerCP, IRDye 600LT, or IRDye 800CW. After washing off excess secondary antibodies, sections were counterstained using 1 µg/ml DAPI (Thermo Fisher Scientific) to visualise cell nuclei. Slides were then coverslipped using Immu-Mount medium (Thermo Fisher Scientific) and imaged using a multispectral epifluorescence microscope (described below). After imaging, tissue bound primary and secondary antibodies were both stripped off the slides after a 5-minute incubation at RT in NewBlot Nitro 5X Stripping buffer (Li-Cor Biosciences) followed by 1-minute additional heat mediated antigen retrieval step in Tris/EDTA buffer. The above processing cycle beginning with re-blocking of tissues in Human BD Fc Blocking solution was repeated and the same sections incubated using additional panels of antibodies of interest (see Supplementary Table 1). The whole process was sequentially repeated up to 10 total cycles to accumulate up to 100 antibody labels for each section.

**Imaging.** Images were acquired from MP-IHC probed whole specimen sections using the Axio Imager.Z2 slide scanning epifluorescence microscope (Zeiss) equipped with a 20X/0.8 Plan-Apochromat (Phase-2) non-immersion objective (Zeiss), a high-resolution ORCA-Flash4.0 sCMOS digital camera (Hamamatsu), a 200 W X-Cite 200DC broadband lamp source (Excelitas Technologies), and 10 customised filter sets (Semrock) optimised to detect the following fluorophores: DAPI, DyLight 405, Alexa Fluor 430, Alexa Fluor 488, Alexa Fluor 546, Alexa Fluor 594, Alexa Fluor 647, PerCP, IRDye 680LT and IRDye 800CW. The specific configurations of the exciter, dichroic, and emitter filters for each fluorophore enabled more narrow bandpass and off-peak excitation/emission properties to minimise filter crosstalk as previously reported[8].

Image tiles (600 × 600 µm viewing area) were individually captured at 0.325 micron/pixel spatial resolution, and the tiles seamlessly stitched into whole specimen images using the ZEN 2 image acquisition and analysis software program (Zeiss), with an appropriate colour table applied to each image channel to either match its emission spectrum or to set a distinguishing colour balance. Pseudocoloured stitched images acquired from all rounds of multiplex IHC staining and imaging were then exported to Adobe Photoshop and overlaid as individual layers to create multicoloured merged composites. The images from each cycle of MP-IHC were aligned manually using the DAPI labelling from each round as a reference.

From the 89 screened antibodies, 25 markers of interest were selected for the spatial protein analysis. Where multiple antibodies were used for the same target antigen or the same antibody was used in different labelling rounds, the antibody/round with the best signal relative to background across all cases was selected for the spatial protein analysis. This was done by comparing the grayscale values for true signal and background for each image. The individual layers corresponding to the 25 markers of interest were exported from Photoshop as separate grayscale TIFF files.

**Spatial protein analysis - differential label analysis.** Bulb images, one per label, were imported into R (https://www.r-project.org/) using the "load.image" function supported in the "imager" library (http://dahtah.github.io/imager/). Images were confirmed to be in grayscale or converted to grayscale using the "grayscale" function. Using a fixed and symmetric bin size (31 × 31 pixels corresponding to a 10 × 10 µm image area), the median value for all non-zero intensities falling in each possible blunt-ended bin per image was calculated and the location for each captured. For bins having no non-zero intensities, a value of "NA" was assigned. These values were then organised by bulb in matrix form with bin location described in rows and label source in columns. Values in these matrices were next multiplied by 100 then rounded to have no decimal place, producing a "pseudo" count matrix per bulb. For each of these matrices, the mean (trim = 0.05) was calculated across bins for all labels then used to generate a Poisson p value per bin per label using the "ppois" function (lower.tail=FALSE). Bins having a p value < 0.05 were coded "1" while those that did not were coded "0". These codings were next interrogated across bulbs in two ways: (1) by-label and (2) by combinations of labels. For by-label interrogation, the number of bins coded "1" were tallied by bulb per label then divided by the total number of bins per bulb, returning the ratio of total bins coded "1". These ratios were then tested for differences across bulbs via ANOVA using the "aov" function under no correction condition and correction condition using the "mt.rawp2adjp" function (proc = "BH"). Post hoc testing was also performed using the "TukeyHSD" function. Labels observed to have an uncorrected ANOVA p value < 0.10 were construed to be differential across bulbs. To assess these differences, bulb-to-bulb relationships were visually inspected by passing the label ratios to the "pheatmap::pheatmap" function to produce a clustered heatmap and similarly passed to the "prcomp" function to produce a covariance-based PCA scatterplot. To interrogate the combinations of labels across bulbs, bins coded "1" or "0" were first concatenated into strings per bin by bulb. The number of times each string was observed per bulb was then tested for differences across bulbs via Kruskal–Wallis test using the "kruskal.test" function under no correction condition and correction condition using the "mt.rawp2adjp"

function (proc = "BH"). Post hoc testing was also performed using the "dunnTest" function. Strings of labels observed to have an uncorrected Kruskal-Wallis p value < 0.05 were construed to be differential across bulbs. To summarise what labels define each of these string, the "heatmap.2" function was used to produce a clustered heat map with the strings described in columns and the labels that compromise them described in rows. To assess how strings are spatially positioned within each bulb, bins representing the differential strings of labels were plotted in image space using colour to denote each unique string.

**Spatial protein analysis—clustering for layer discovery**. The "pseudo" count matrices generated per bulb (see "Differential Label Analysis") were processed in R (https://www.r-project.org/) using functions supported in the "Seurat" library (https://www.rdocumentation.org/packages/Seurat/versions/3.1.4). Specifically, each matrix was recast as a "Seurat" object using the "CreateSeurateObject" function. Each of these objects were then separately pushed through the following functions under default settings: "SCTransform", "RunPCA", "ElbowPlot", "RunUMAP", "FindNeighbors", "FindClusters". For the "RunUMAP" and "Find-Neighbors" functions, the maximum number of dimensions passed was set equal to the number of principal components indicated by the "ElbowPlot" function to explain > 1 SD. Clusters of bins returned from the "FindClusters" function were construed to represent "layers" present within each bulb, respectively, with each "layer" consisting of a unique combination of labels and/or "pseudo" counts for those labels. To compare and contrast the combinations of labels and "pseudo" count level for those labels, the "pheatmap::pheatmap" and "DotPlot" functions were used. While single label inspection across layers for a bulb was accomplished using the "VlnPlot" and "RidgePlot" functions. To assess how layers are spatially positioned within each bulb, bins were plotted in image space using colour to denote layer membership. The spatial protein analysis utilized the computational resources of the NIH HPC Biowulf cluster (http://hpc.nih.gov), specifically a compute node with 2 processors, 500GB memory and local drive storage of 500GB.

**Reporting summary**. Further information on research design is available in the Nature Research Reporting Summary linked to this article.

## Data availability

The image dataset is publicly hosted at https://doi.org/10.17608/k6.auckland.14920233. All data that support the findings of this study are available from the corresponding author upon reasonable request.

## Code availability

The code used for this analysis is publicly hosted at https://doi.org/10.17608/k6.auckland.17132603.v1. Rstudio version 3.6.1 was used for the spatial protein analysis. Clustering functions supported in the "Seurat" library (https://www.rdocumentation.org/packages/Seurat/versions/3.1.4). The spatial protein analysis utilized the computational resources of the NIH HPC Biowulf cluster (http://hpc.nih.gov), specifically a compute node with 2 processors, 500GB memory and local drive storage of 500GB.

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

## Acknowledgements
This research was supported by the Intramural Research Program of the NIH, NINDS. The authors would like to thank the following organisations for funding: HCM is a Health Education Trust Postdoctoral Research Fellow and this research was funded by the Health Education Trust. BH is funded by a Brain Research New Zealand doctoral scholarship. BVD is funded by the Michael J Fox Foundation (16420) and the NeuroResearch Charitable Trust. Special thanks go to the Neurological Foundation of New Zealand for funding the Neurological Foundation Human Brain Bank at the University of Auckland and to Marika Ezes and all technical staff involved in the collection and processing of the human brain tissue at the Centre for Brain Research and the Human Anatomy Labs.

## Author contributions
H.C.M., A.K., and D.M. contributed to the conception and design of the experiments. Tissue processing was performed by H.C.M, B.V.D, R.L.M., and M.A.C. MP-IHC labelling was carried out by A.S. Microscopy was carried out by D.M. Image analysis was developed by K.J. and H.C.M. with input from P.V.A. Data interpretation was carried out by H.C.M,. K.J., and B.H. with critical revision by A.K. and M.A.C. The manuscript was prepared by H.C.M. with critical revision by D.M., A.K., and M.A.C. and feedback from all authors. All authors have approved the final manuscript.

## Competing interests
The authors declare no competing interests.
