## [Transparent Peer Review File · Communications Biology]

Reviewers' comments:

Reviewer #1 (Remarks to the Author):

The manuscript by Murray et al, shows the use of high-plex immunofluorescence labeling in a single tissue section to allow direct comparison of many markers. Up to 100 markers have been used in human olfactory bulb sections from normal patients, with Alzheimer's and Parkinson's disease. Their results present a complete immunohistological characterization of human OB anatomy and a summary of the markers differentially expressed in AD and PD. The MP-IHC approach has great potential and can be useful for a wide range of researchers.

There are some considerations I believe need to be addressed to improve the data presentation and verify the conclusions of this study.

1. I don't think the term "proteomics" is appropriate for this kind of study. Please modify.

2. The "normal" group nomenclature is not correct, since it includes aging and surely other pathologies. It would be convenient to modify by "control" group.

3. Methods

a. In Table 2 include the data of the neuropathological stages of the AD and PD cases.

b. What controls have been carried out in the immunohistochemistry?

c. Different antibodies have been used for the same antigen, what has been the criterion for selecting one over the others? For example,

NeuN Millipore ABN90P (used in rounds 2, 3 and 10) is not selected for study; Neu N Millipore ABN91*, only used in round 5.

Beta Amyloid (6E10)*, round 2; Beta Amyloid (6F3D) round 3; Beta Amyloid (MOAB-) round 5 and 10
Calbindin ... among others,

please explain.

4. Results

a. Efficient neurochemical characterisation of the human OB using multiplex IHC. To make a cytoarchitectural map it would be necessary to use different sections of the OB, the fact of using a single middle section (aesthetically more beautiful) biases the study. Regarding the comparison between controls, ad and pd: a single section is not enough to obtain reliable comparative results. The results would be more robust if the layers were measured.

b. Spatial proteomics detects differential abundance of antibody labelling between normal and disease groups. Line 222 "One PD and one normal case clustered with the AD cases? Why? Are these cases possible outliers that should be removed from the study? Do they stray from their original study group?"

c. Sub-clustering of tau-positive bins identifies markers associated with pathology. Line 231 "To further investigate the spatial and neurochemical signature of disease". I assume it refers to AD, indicate correctly.

5. Discussion

a. Line 265. "we have presented the first comprehensive neurochemical 265 characterisation of the human OB." This is not the first study, since there are others in the previous literature. Since the 1990s, studies of this type have been carried out, although with different approaches (Smith et al 1990, Ohm et al 1991, Smith et al 1993). I suggest that the authors review the bibliography and discuss it.

b. Line 292. "Tau, beta-amyloid and α -synuclein aggregates are hypothesised to spread from the AON to downstream brain regions as the disease progresses."

Regarding the pathological proteins (alpha-synuclein, beta amyloid and tau) and their expression and distribution in each of the sections shown (supplementary figure 6), some doubts arise that should be resolved:

- Pathological proteins in AD and PD are usually widely shown in bulbar AON, but this is not what is observed in the figure. Of the AD cases, there is hardly any beta amyloid in the OB, while in some of them (AZ84) there is a large amount of alpha (is there an associated pathology? If so, show in table 2). The AZ99 case, according to its bibliographic reference (10) has Braak's stage V-VI, which would lead to think that we would find a large amount of pathological protein in OB, which does not occur,

is it due to the technique? Have you performed antibody controls to verify that the labeling is correct?
- Control cases are 63 to 94 years old. Except for H250, none of them show any of the pathological proteins, do you consider it normal for such an aged OB to be completely clean of each and every one of these proteins? Does not affect aging? Please discuss these results.

I insist on the need to give some kind of verification / control of the immunofluorescent technique, which helps to generate confidence in what is shown.

c. Line 309. "Our analysis showed a significant increase in tau and GFAP in the AD OB, which is indicative of tau neuropathology and astrogliosis as previously described^{10,26,30}". Check the bibliography you cite and discuss correctly, you are comparing AD with PD.

d. Line 316. "In contrast, our MP-IHC and spatial proteomics approach provide an efficient pipeline to screen for changes on a single tissue section." Note that the OB is a three-dimensional structure, the layers of which can vary throughout it, a study in a single section is not comparable with a study of the complete structure. Please weigh the pros and cons of your technique and discuss them.

Reviewer #2 (Remarks to the Author):

In this work, Murray et al. have demonstrated high-throughput multiplexed imaging through iterated staining, imaging, and removal of antibodies. Through the use of fluorophores and filter sets that minimize crosstalk between the fluorophores, the authors have successfully used ten fluorophores in a single cycle, thereby greatly reducing the total process time and tissue damage. Using this approach, the authors have studied the molecular heterogeneity of human brains, especially in AD and PD patients. The method demonstrated in this work (even though it was published in a separate paper) and the spatial proteomics analysis pipeline the authors have built would be helpful for researchers who study brain disorders. Here are questions I would like to ask before the acceptance of this manuscript.

1. Recently, various antibody stripping techniques have been developed. Why did the authors use NewBlot Nitro 5X stripping buffer (Li-Cor Biosciences) to remove antibodies from specimens? What is the major advantage of using this buffer compared to previously reported reagents?

2. Related to the above question, previous studies have reported that several antibodies with high binding affinity were not completely removed after the antibody stripping process. Did the authors observe a similar phenomenon?

3. In the same authors' previous work (Nat. Commun. 12:1550 2021), they noted that around 5% of the spectral bleed-through was observed even with the optimized fluorophores and filter sets. To minimize the signal crosstalk, the determination of a pair of fluorophores and antibodies would be critical, I believe. Did the authors use any specific selection criteria for their fluorophore and antibody pairs?

4. Unlike the vast majority of papers related to clustering analysis, the authors achieved spatial proteomic analysis through approaches for spatial transcriptomics and single-cell genomics analysis. Why did the authors choose a different strategy than was used in previous papers? Is pixel bin segmentation more robust than cell segmentation? Can potential users achieve accurate clustering results through pixel bin segmentation when more than two cell types exist in a single pixel bin?

5. The measurement of fluorescence intensity in each pixel bin unit is mainly based on creating a binary image plane and thresholding the Z-score > 0.05 . It would be useful to know whether the chosen threshold condition is optimized and works well in all the cases or if it should be optimized for every experiment.

6. As a follow-up question, the authors mentioned that the pixel bin size can be adjusted to modify the resolution (line 284 in the main text). If so, can this proposed method be utilized to visualize more detailed structures?

Reviewer #3 (Remarks to the Author):

The manuscript by Dr. Murray and colleagues describes how a multiplexed fluorescence-based immunohistochemistry (MP-IHC) method applied to the human olfactory bulb can establish different "proteomic" pattern among normal, Alzheimer and Parkinson cases. The manuscript is technically sound, relevant and very interesting for a broad audience. Several points, however, could improve it.

Major concerns

-the concept of "spatial proteomics" herein used is confusing since proteomic approaches usually uses mass spectrometry and particularly when applied on tissue, reader assumes Matrix-assisted laser desorption/ionization (MALDI) time-of-flight (TOF) imaging mass spectrometry (IMS). Please, clarify this point.

-Authors claims that this method is unbiased, but a single section is used, which is a bias itself. Only central and aesthetically suitable sections are used. Any stereological approach assumes volume samples selecting several sections randomly. Please correct.

-Authors establish AD, PD and "normal" group. This is incorrect. The third group should be named non-diseased or better Non-AD/PD. The olfactory bulb of "normal" people above 60 years very likely display proteinopathies.

-In figures 5 and 6 tau is used as a marker to discriminate AD group. Why alpha-synuclein is not used to do the same in PD group?

-Interestingly and related to the above point, discuss co-pathologies should be much interesting using this technique. Please, include in the discussion.

-Also in the discussion, include more previous literature on different markers in the human OB.

Reviewer #4 (Remarks to the Author):

This is a well presented study of human olfactory bulbs obtained post mortem from patients with Alzheimer's, Parkinson's and controls with no known neurodegenerative disease. The main novelty of the study is the high-resolution neuroanatomy and pathology findings obtained using multiplex immunofluorescence and a spatial proteomic analysis on single tissue sections. The technique has been previously used by the authors in rodent tissue (Ref 7, Maric et al.) but this is the first application to human brain tissue. The results are fascinating and this technique appears to offer some new insights into the structure of the human olfactory bulb and the pathologies that affect it.

As a neuropathologist, I just have a couple of questions:

1. Post mortem human olfactory bulb tissue is notoriously fragile and the bulbs are often damaged on removal. The specimens used here appear to be intact but how did the investigators manage to maintain the tissue structural integrity through 10 cycles of their multiplex immunofluorescence involving multiple rinsing and coverslip removal steps?

2. While this is largely a technique paper I think it would be strengthened if the discussion on the findings regarding the novel anatomical observations and the anatomical specificity of the tau and α -synuclein pathology was expanded. There is a wealth of data in the supplementary material mapping the pathology in the different cases but a summary figure illustrating the differential distribution of the pathology in AD and PD cases would help the reader to see this. Can the authors speculate a little on why these differences occur?

Reviewer #1 (Remarks to the Author):

The manuscript by Murray et al, shows the use of high-plex immunofluorescence labeling in a single tissue section to allow direct comparison of many markers. Up to 100 markers have been used in human olfactory bulb sections from normal patients, with Alzheimer's and Parkinson's disease. Their results present a complete immunohistological characterization of human OB anatomy and a summary of the markers differentially expressed in AD and PD. The MP-IHC approach has great potential and can be useful for a wide range of researchers.

There are some considerations I believe need to be addressed to improve the data presentation and verify the conclusions of this study.

We would like to thank reviewer 1 for comprehensive and thoughtful review of our manuscript. We appreciate that the reviewer recognises the potential of the approach for a wide range of researchers. We have carefully considered the reviewer's comments, responded to all raised points as detailed below, and revised the manuscript and figures accordingly.

1. I don't think the term "proteomics" is appropriate for this kind of study. Please modify.

The authors have carefully considered this comment, along with reviewer 3, comment 1, and agree that the term proteomics has traditionally been applied to techniques such as mass-spectrometry where proteins are directly measured. As our multiplex IHC approach assesses proteins indirectly using antibody labelling we agree that in this context, it would be more appropriate to refer to our analysis as 'spatial protein analysis' and have amended this terminology throughout the manuscript and figures.

2. The "normal" group nomenclature is not correct, since it includes aging and surely other pathologies. It would be convenient to modify by "control" group.

In previous publications we have used the terminology 'neurologically normal' to describe our aged comparison group. This is because the term 'control' implies there was an experimental manipulation applied to the other groups. As our studies of human brain tissue are observational, we do not believe the term 'control' is appropriate. However, we appreciate the reviewer's comment that aging and the potential for other pathologies are not well represented with the term 'normal' and have modified the nomenclature to "No significant pathology" (NSP) throughout the manuscript and figures.

3. Methods

a. In Table 2 include the data of the neuropathological stages of the AD and PD cases.

As requested, we have added the tau Braak stage, beta-amyloid Thal stage and the pathologist's description of α -synuclein pathology for all cases (where available) in Table 2. Neuropathological Braak staging of the PD cases is not available.

We also added a note that only the bulbs were obtained for OFB6A and OFB57 and no pathological staging was performed on the rest of the brain.

b. What controls have been carried out in the immunohistochemistry?

All antibodies used for this study have been previously validated for use in formalin-fixed, paraffin-embedded rodent and human brain tissues as cited in our previous publications (Maric et al., 2021; Murray et al., 2020; Stevenson et al., 2020) and the extensive literature referenced by the RRID for each antibody (Supplementary Table 1) in The Antibody Registry (<https://antibodyregistry.org/>). This gave us a clear expectation for the high-fidelity labelling for each antibody.

The antibody labelling was also empirically validated in this study based on the internally controlled experimental design of the multiplex IHC compatible antibody panels applied in each tissue staining round, whereby, each antibody used in a given staining round predominantly targeted a spatially distinct sub-cellular location or tissue structure, with each primary antibody selected from a different animal host and/or immunoglobulin class/subclass, visualized using an appropriate secondary antibody conjugated to a different spectrally non-overlapping fluorophore, and the resulting fluorescence signal from each antibody imaged in a different spectrally non-overlapping channel, with minimal spectral crosstalk, as previously reported (Maric et al., 2021). This way, we internally controlled for cross-reactivity between antibodies and any residual spectral crosstalk between fluorescence signals in each imaging channel for each staining round. Furthermore, in each subsequent labelling round, we shuffled the antibody panel to target spatially unrelated proteins and/or structures which were imaged in the previous round by switching the animal host and immunoglobulin class/subclass of the primary and secondary antibodies for each imaging channel, thus allowing us to confirm that antibody labelling in each previous round has been effectively removed by the antibody stripping procedure performed prior to each subsequent round of labelling. To demonstrate this empirical validation, we have added a new Supplementary Figure 1, which shows the labelling for every antibody used in this study. The results show no significant cross-reactivity between antibodies nor spectral crosstalk between fluorescence signals in each imaging channel for each staining round. Unsurprisingly, some antibodies we included in the screening panels did not produce a compelling signal against the expected target and were thus not included in the quantitative analysis.

We have added additional comments regarding this empirical validation to the methods section (lines 430-435).

New Supplementary Figure 1. Summary of antibody labelling and empirical validation of MP-IHC

*c. Different antibodies have been used for the same antigen, what has been the criterion for selecting one over the others? For example, NeuN Millipore ABN90P (used in rounds 2, 3 and 10) is not selected for study; Neu N Millipore ABN91 *, only used in round 5. Beta Amyloid (6E10) *, round 2; Beta Amyloid (6F3D) round 3; Beta Amyloid (MOAB-) round 5 and 10 Calbindin ... among others, please explain.*

Multiple antibodies were used for the same target antigen since some antibodies (particularly polyclonals) exhibited significant non-specific background labelling in different tissue samples and we wanted to empirically select the antibody with optimum labelling for the spatial analysis to be

consistent for all tissue samples screened. Where multiple antibodies were used for the same target protein, we selected the antibody that consistently showed the best signal compared to background across all cases. This was done by comparing the grayscale values for true signal and tissue background signal for each image. In this regard, chicken polyclonal IgY anti-NeuN (ABN91) antibody had higher signal and lower background compared polyclonal guinea pig IgG anti-NeuN (ABN90P) antibody in some OB tissue sections. However, ABN90P antibody was used in multiple rounds since the guinea pig host species was immunocompatible with the rest of the panel antibodies and helped interpret or empirically validate the labelling of other antibodies in those rounds prior to image alignment.

For the antibodies that were repeated across several rounds and were included in the spatial analysis (histones, tomato lectin, MAP2, tau AT8), we selected the best image for analysis based on the signal to background criteria, by comparing the grayscale values for true signal and background in each image.

We have added additional clarification on this to the methods section in lines 475-481.

4. Results

a. Efficient neurochemical characterisation of the human OB using multiplex IHC. To make a cytoarchitectural map it would be necessary to use different sections of the OB, the fact of using a single middle section (aesthetically more beautiful) biases the study. Regarding the comparison between controls, ad and pd: a single section is not enough to obtain reliable comparative results. The results would be more robust if the layers were measured.

We agree that multiple sections would be required to make a full cytoarchitectural map of the human OB. Our decision to use a single section from multiple cases was based on the main aim of this manuscript to demonstrate the feasibility of the multiplex IHC technique and spatial analysis for exploring the laminar distribution of the human bulb, which has had limited neurochemical characterisation in the literature published thus far. As bulb structure is highly variable between individual cases, we chose to explore a single section across more cases to ensure our approach identified layer signatures consistently despite variable bulb shape and size. This is consistent with previous technique papers such as

We agree that additional OB sections would be required to validate the results for differentially expressed markers between groups and this data should be interpreted with the caveat that only one section was analysed. We have added these comments to the results (line 213-217) and discussion (line 329-334).

b. Spatial proteomics detects differential abundance of antibody labelling between normal and disease groups. Line 222 "One PD and one normal case clustered with the AD cases? Why? Are these cases possible outliers that should be removed from the study? Do they stray from their original study group?"

The cases that clustered with the AD cases (Figure 5b) are PD58 and H190. We have added annotations to Figure 2b to clarify the position of each case. These two cases illustrate the complexity of pathology in ageing and disease. After obtaining additional pathology information as requested in comment 3a, we noted that PD58 has intermediate Alzheimer's pathology despite a clinical diagnosis of Parkinson's disease and pathological diagnosis of Neocortical diffuse Lewy body

disease (Table 2, now revised). Therefore, the clustering of PD58 with the AD cases does align with the overall brain pathology. Furthermore, it clustered very closely with AZ84 which is an AD case with neocortical alpha-synuclein pathology. We have added comments to the discussion regarding these findings (Line 349-357)

H190 has no pathology that suggests it should be excluded from the no significant pathology group. Instead, we believe this case clustered with the AD group because the expression of calretinin, S100, synaptophysin and TH (differentially expressed markers, Figure 5A) more closely matched those of the AD cases.

We believe these outliers in the clustering analysis are a good demonstration of how the multiplex IHC technique could be used to cluster cases based on anatomical features of the tissue, producing a more nuanced comparison of cases with comorbid pathology.

Figure 5 Revised. Annotations have been added to (b) to clarify the position of each case

c. Sub-clustering of tau-positive bins identifies markers associated with pathology. Line 231 "To further investigate the spatial and neurochemical signature of disease". I assume it refers to AD, indicate correctly.

We thank the reviewer for making us aware of this and have modified the sub-heading and opening sentence of the paragraph accordingly.

5. Discussion

a. Line 265. *"we have presented the first comprehensive neurochemical 265 characterisation of the human OB." This is not the first study, since there are others in the previous literature. Since the 1990s, studies of this type have been carried out, although with different approaches (Smith et al 1990, Ohm et al 1991, Smith et al 1993). I suggest that the authors review the bibliography and discuss it.*

We agree that previous studies have provided neurochemical characterisation of many markers in the human bulb and appreciate that we have not provided the first report for many of the markers presented, although we maintain that this is a more comprehensive overview of human bulb anatomy than has been previously presented. We initially refrained from a detailed discussion of previous bulb anatomy literature as the manuscript is primarily intended to demonstrate the multiplex IHC and spatial analysis approach. However, we appreciate the requests from reviewer 1 and reviewer 3 to add a more comprehensive bibliography that would support the results we have presented and have therefore added additional comments and references to previous neurochemical studies of the human olfactory bulb in lines 275-288.

b. Line 292. *"Tau, beta-amyloid and α -synuclein aggregates are hypothesised to spread from the AON to downstream brain regions as the disease progresses."*

Regarding the pathological proteins (alpha-synuclein, beta amyloid and tau) and their expression and distribution in each of the sections shown (supplementary figure 6), some doubts arise that should be resolved:

- Pathological proteins in AD and PD are usually widely shown in bulbar AON, but this is not what is observed in the figure. Of the AD cases, there is hardly any beta amyloid in the OB, while in some of them (AZ84) there is a large amount of alpha (is there an associated pathology? If so, show in table 2). The AZ99 case, according to its bibliographic reference (10) has Braak's stage V-VI, which would lead to think that we would find a large amount of pathological protein in OB, which does not occur, is it due to the technique? Have you performed antibody controls to verify that the labeling is correct?

Regarding the low amount of beta-amyloid in the AD cases, this is consistent with what we observed in our previous paper that performed fluorescent IHC for amyloid and tau on these same cases (Murray et al, 2020, see graphs below). In this previous paper we conducted a semi-quantitative assessment of beta-amyloid and tau load in the AON and found that amyloid load ('D' in the figure below) was much less abundant than tau ('C' in the figure below) in the AON. There was also considerable variability in the amount of tau and beta amyloid in the AON between cases and the load did not correlate with Braak staging (see Murray et al, 2020). Therefore, we do not expect that the AD cases with higher Braak stage should have more tau pathology.

Figure 3 from Murray et al, 2020.

Regarding the location of aggregates in the bulbar AON. We maintain that Supplementary Figures 3 and 6 (now revised to supplementary figures 4 and 7) do show that the pathological proteins are predominantly located in the AON (predominantly tau for AD and alpha-synuclein for PD), keeping in mind that the AON has a very different shape and structure for each case. It should also be noted that supplementary Figure 6 (now 7) shows the binary segmentation for each marker, that is, the pixel bins that were determined to be positive for labelling of each marker after thresholding, whereby the Poisson p-value was calculated per bin per label per section and bins having a p-value < 0.05 were coded "1" while those that did not were coded "0". Considering the beta amyloid and alpha-synuclein labelling was sparse and the objects are very small (diffuse amyloid and small lewy neurites), the binning and thresholding approach has resulted in some labelling being excluded and therefore fewer positive bins in the spatial slide maps in now supplementary Figure 7.

For AZ84, the pathology report indicates that this case has neocortical Lewy body disease, suggesting the presence of alpha synuclein throughout the brain. We have added this additional pathology information to Table 2.

Lastly, we do not believe the technique has reduced the amount of labelling for pathological proteins as the labelling is comparable to single round labelling that we have performed on these cases. Illustration of this is provided in the OB images below, with comparisons of tau and beta amyloid for two AD cases and alpha synuclein for two PD cases. These images also illustrate that the tau and alpha synuclein aggregates are predominantly located in the AON and the amount of pathology does vary between cases.

AZ90 Section 214 Single round

DAPI Amyloid Tau

AZ90 Section 242 Multiplex

DAPI Amyloid Tau

AZ99 Section 139 Single round

DAPI Amyloid Tau

AZ99 Section 140 Multiplex

DAPI Amyloid Tau

PD52 Section 206 Single round

DAPI Alpha synuclein

PD52 Section 242 Multiplex

DAPI Alpha synuclein

PD77 Section 145 Single round

DAPI Alpha synuclein

PD77 Section 197 Multiplex

DAPI Alpha synuclein

- Control cases are 63 to 94 years old. Except for H250, none of them show any of the pathological proteins, do you consider it normal for such an aged OB to be completely clean of each and every one of these proteins? Does not affect aging? Please discuss these results.

As explained above, the diagrams in Supplementary Figure 6 (now 7) show the pixel bins that were determined to be positive for labelling of each marker after thresholding. As can be seen in Supplementary Figure 3 (now 4), there is a very small amount of tau observed in H250 and H190; however, the thresholding approach we have used has resulted in no positive bins for these labels in Supplementary Figure 6 (now 7). As the priority for this manuscript is to demonstrate the spatial analysis of multiplex IHC labelling to screen for differences between groups, we believe it was important to apply a consistent thresholding approach to all sections in the analysis.

Furthermore, the low amount of pathological proteins in the normal bulbs is consistent with what we observed in our previous paper (Murray et al, 2020), which had more control cases. As seen in Figure 3C and 3D from Murray et al, 2020 above, the normal bulbs in our study had no pathological aggregates or have very little relative to AD or PD.

I insist on the need to give some kind of verification / control of the immunofluorescent technique, which helps to generate confidence in what is shown.

As per our response to comment 3b above, we have added a new Supplementary Figure 1 to demonstrate the empirical validation of the antibodies used in this study, as was also presented in our previous paper (Maric et al., 2021). It would be extraordinarily expensive in terms of reagents and tissue to perform individual validation of all 89 antibodies across 10 rounds of labelling. We believe this empirical validation with internal controls allows verification of the effectiveness of the antibody stripping procedure, consistency of labelling across the rounds, absence of cross-reactivity between labelling rounds and absence of spectral crosstalk between imaging channels, all of which supports our confidence in the results. We have added additional comments regarding this empirical validation to the methods section (lines 430-435).

c. Line 309. "Our analysis showed a significant increase in tau and GFAP in the AD OB, which is indicative of tau neuropathology and astrogliosis as previously described10,26,30". Check the bibliography you cite and discuss correctly, you are comparing AD with PD.

We appreciate the reviewer picking up on this discrepancy. The sentence should read "Our analysis showed a significant increase in tau and GFAP in the AD **and PD** OB, which is indicative of tau neuropathology and astrogliosis as previously described". This statement relates to Figure 5C. We have corrected it in the discussion.

d. Line 316. "In contrast, our MP-IHC and spatial proteomics approach provide an efficient pipeline to screen for changes on a single tissue section." Note that the OB is a three-dimensional structure, the layers of which can vary throughout it, a study in a single section is not comparable with a study of the complete structure. Please weigh the pros and cons of your technique and discuss them.

As per our response to comment 4a, we agree that additional sections would be required to validate the results for differentially expressed markers between groups and the data in this study should be interpreted with the caveat that only one section was analysed. We have added comments to the results (line 213-217) and discussion (line 329-333).

Reviewer #2 (Remarks to the Author):

In this work, Murray et al. have demonstrated high-throughput multiplexed imaging through iterated staining, imaging, and removal of antibodies. Through the use of fluorophores and filter sets that minimize crosstalk between the fluorophores, the authors have successfully used ten fluorophores in a single cycle, thereby greatly reducing the total process time and tissue damage. Using this approach, the authors have studied the molecular heterogeneity of human brains, especially in AD and PD patients. The method demonstrated in this work (even though it was published in a separate paper) and the spatial proteomics analysis pipeline the authors have built would be helpful for researchers who study brain disorders. Here are questions I would like to ask before the acceptance of this manuscript.

We would like to thank reviewer 2 for the comments recognising the value of this labelling and analysis approach. Immunohistochemistry on human brain tissue from neurodegenerative disease patients is particularly challenging and the method we present will allow more information to be obtained from this precious resource. We have carefully considered the comments, responded to all points below, and revised the manuscript accordingly.

1. Recently, various antibody stripping techniques have been developed. Why did the authors use NewBlot Nitro 5X stripping buffer (Li-Cor Biosciences) to remove antibodies from specimens? What is the major advantage of using this buffer compared to previously reported reagents?

We tested many previously reported reagents for the antibody stripping, typically applied to Western blot applications. We found the NewBlot Nitro 5X stripping buffer and subsequent heat-mediated antigen retrieval in TrisEDTA pH9.0 was the most consistent in terms of stripping many different antibodies and did not degrade the tissue or cause the tissue to lift from the slide, despite multiple rounds of antibody stripping and relabelling. Another advantage of using the aforementioned stripping buffer is that it is commercially available and therefore widely accessible and consistent.

2. Related to the above question, previous studies have reported that several antibodies with high binding affinity were not completely removed after the antibody stripping process. Did the authors observe a similar phenomenon?

Both primary and secondary antibodies used in this study were effectively removed after each antibody stripping process. Our antibody stripping protocol actually involves two steps, by first applying the full strength (neat) NewBlot Nitro 5x stripping buffer directly on the tissue section, followed by TrisEDTA heat-mediated antigen retrieval/antibody stripping step carried out on each tissue sample prior to each additional round of antibody labelling. To demonstrate the utility of our approach, and also in response to Reviewer 1 comment 3b above, we have added a new

Supplementary Figure 1, which clearly shows the effectiveness of antibody stripping in each additional round of staining by the absence of target signal in each imaged channel acquired from the previous round of antibody labelling.

A further consideration of the effectiveness of removal of residual signals potentially left over after antibody stripping steps is that our multiplex IHC protocol uses secondary antibody amplification of each primary antibody in each new round of staining. Therefore, once the primary and secondary antibodies have been effectively stripped off the tissue after each staining/imaging round, the subsequent primary antibody labelling of new antigenic targets and secondary antibody amplification to visualize those targets in the new round of labelling results in a significantly higher signal than any signal potentially remaining from any unstripped antibodies from the previous round of labelling. This means that when the new round of labelling is imaged using an appropriate light and camera exposure setting on the microscope, any residual labelling potentially remaining from the previous round does not produce enough signal to be captured in the new round of imaging.

3. In the same authors' previous work (Nat. Commun. 12:1550 2021), they noted that around 5% of the spectral bleed-through was observed even with the optimized fluorophores and filter sets. To minimize the signal crosstalk, the determination of a pair of fluorophores and antibodies would be critical, I believe. Did the authors use any specific selection criteria for their fluorophore and antibody pairs?

Yes, as mentioned in response to Reviewer 1, comment 3b, the design of the antibody-fluorophore panels was critical for empirical validation of the labelling and minimization of spectral crosstalk. The primary consideration for selection criteria of fluorophore-antibody pairs was that each primary antibody used in a given staining round was selected to predominantly target a spatially distinct sub-cellular location or tissue structure, with each antibody sourced from a different animal host and/or immunoglobulin class/subclass to minimize cross-reactivity, and each immunoreaction visualized using an appropriate secondary antibody conjugated to a different spectrally non-overlapping fluorophore, with the resulting fluorescence signal from each antibody imaged in a different spectrally non-overlapping channel, with minimal signal crosstalk, as demonstrated in Supplementary Figure 1.

We have added additional comments regarding this panel construction and empirical validation to the methods section (line 430-435).

4. Unlike the vast majority of papers related to clustering analysis, the authors achieved spatial proteomic analysis through approaches for spatial transcriptomics and single-cell genomics analysis. Why did the authors choose a different strategy than was used in previous papers? Is pixel bin segmentation more robust than cell segmentation? Can potential users achieve accurate clustering results through pixel bin segmentation when more than two cell types exist in a single pixel bin?

Cellular segmentation was particularly difficult considering our aim was a cytoarchitectural analysis rather than a single-cell approach. Most of our markers were not cellular as we wanted to assess complex tissue features such as blood vessels, white matter and the bulb layers. We used pixel bin

segmentation as it required minimal user input and allowed us to investigate a range of tissue features to that could contribute to layer delineation.

Single cell or single object segmentation would be more appropriate for specific biological questions such as cell phenotyping, and we are currently working on a pipeline for a cell segmentation approach. However, we believe the pixel bin analysis presented here is a novel approach to screen multiplex image datasets for interesting tissue features with minimal user input. Our demonstration of its use for investigating differentially expressed markers also showed that the pixel bin labelling signature across a section could be a more nuanced approach for classifying tissue into disease groups.

We have added additional comments on the use of this analysis to the discussion in lines 300-304.

5. The measurement of fluorescence intensity in each pixel bin unit is mainly based on creating a binary image plane and thresholding the Z-score > 0.05 . It would be useful to know whether the chosen threshold condition is optimized and works well in all the cases or if it should be optimized for every experiment.

The Z-score threshold was chosen because it provided an accurate representation of all the markers included in the analysis. As the intent was to provide minimal user input and make the approach relevant to observers who may not be an expert in the expected distribution for every marker, we chose not to define a different threshold for each marker. The threshold is relative to the distribution of pixel intensities for each marker and each case, so it is internally normalised and thus works well across different sections/cases. We expect this threshold approach would be appropriate for screening multiplex data sets for interesting tissue features with minimal user input, but there may be some limitations where very small objects might not be detected due to their size relative to the size of the pixel bin. We noted that very small and infrequent structures like alpha-synuclein positive Lewy neurites were not captured using this thresholding.

6. As a follow-up question, the authors mentioned that the pixel bin size can be adjusted to modify the resolution (line 284 in the main text). If so, can this proposed method be utilized to visualize more detailed structures?

Yes, smaller pixel bins would allow this analysis to detect finer structures if required. It can be customised based on the investigator's requirements and biological question. However due to the extremely large number of pixel bins in a single image of the size and resolution we present here, reducing the pixel bin size would significantly increase the number of bins and therefore require much more computing power for the analysis. Therefore, if more detailed structures were to be investigated using smaller pixel bin sizes, a smaller image area should be used.

Reviewer #3 (Remarks to the Author):

The manuscript by Dr. Murray and colleagues describes how a multiplexed fluorescence-based immunohistochemistry (MP-IHC) method applied to the human olfactory bulb can establish different

"proteomic" pattern among normal, Alzheimer and Parkinson cases. The manuscript is technically sound, relevant and very interesting for a broad audience. Several points, however, could improve it.

We would like to thank reviewer 3 for the comments and particularly for recognising the interest of this manuscript to a broad audience. We have carefully considered the comments, responded to all points, and revised the manuscript accordingly.

Major concerns

-the concept of "spatial proteomics" herein used is confusing since proteomic approaches usually uses mass spectrometry and particularly when applied on tissue, reader assumes Matrix-assisted laser desorption/ionization (MALDI) time-of-flight (TOF) imaging mass spectrometry (IMS). Please, clarify this point.

We have carefully considered this comment as it was also raised by reviewer #1 and agree that the term proteomics has traditionally been applied to techniques such as mass-spec where proteins are directly measured. As our multiplex IHC approach assesses proteins indirectly using antibody labelling we agree that in this context, it would be more appropriate to refer to our analysis as 'spatial protein analysis' and have amended this terminology throughout the manuscript and figures.

-Authors claims that this method is unbiased, but a single section is used, which is a bias itself. Only central and aesthetically suitable sections are used. Any stereological approach assumes volume samples selecting several sections randomly. Please correct.

We agree with the reviewer that the term 'unbiased' is not appropriate due to the selection of a single section for the analysis. We have removed this term throughout the manuscript and replaced it with the term 'unsupervised'.

-Authors establish AD, PD and "normal" group. This is incorrect. The third group should be named non-diseased or better Non-AD/PD. The olfactory bulb of "normal" people above 60 years very likely display proteinopathies.

We have carefully considered this comment as it was also raised by reviewer #1. As noted for reviewer #1, comment 2: In previous publications we have used the terminology 'neurologically normal' to describe our aged comparison group. This is because the term 'control' implies there was an experimental manipulation applied to the other groups. As our studies of human brain tissue are observational, we do not believe the term 'control' is appropriate. However, we appreciate the reviewer's comment that aging and the potential for other pathologies are not well represented with the term 'normal' and have modified the nomenclature to "No significant pathology" (NSP) throughout the manuscript and figures.

-In figures 5 and 6 tau is used as a marker to discriminate AD group. Why alpha-synuclein is not used to do the same in PD group?

As the primary intent of the manuscript is to demonstrate the application of the multiplex IHC and analysis approach, we aimed to show how the spatial protein analysis could be used to investigate

co-labelling/spatial proximity of markers. We chose to investigate co-labelling of markers with tau in this example as the analysis found tau to be differentially expressed in AD.

Alpha-synuclein was not found to be differentially expressed in the analysis, presumably because the PD cases we examined had variable amounts of alpha-synuclein labelling and the AON contains more Lewy neurites than Lewy bodies, and their small size relative to the pixel bin size resulted in few bins being included after the thresholding process.

-Interestingly and related to the above point, discuss co-pathologies should be much interesting using this technique. Please, include in the discussion.

We agree with the reviewer that investigating co-pathologies with MP-IHC and spatial protein analysis would be an interesting and important application. We have added a section to the discussion (line 349-373) to detail previous evidence of co-pathologies in the human bulb and how the pipeline we present would be an efficient and useful tool to understand this aspect of neurodegenerative disease.

-Also in the discussion, include more previous literature on different markers in the human OB.

As per our response to reviewer 1, comment 5a, we have included additional discussion and references to previous neurochemical studies of the human olfactory bulb in lines 275-283.

Reviewer #4 (Remarks to the Author):

This is a well presented study of human olfactory bulbs obtained post mortem from patients with Alzheimer's, Parkinson's and controls with no known neurodegenerative disease. The main novelty of the study is the high-resolution neuroanatomy and pathology findings obtained using multiplex immunofluorescence and a spatial proteomic analysis on single tissue sections. The technique has been previously used by the authors in rodent tissue (Ref 7, Maric et al.) but this is the first application to human brain tissue. The results are fascinating and this technique appears to offer some new insights into the structure of the human olfactory bulb and the pathologies that affect it.

We would like to thank reviewer 3 for the comments and particularly for acknowledging the novelty of this technique applied to human brain tissue and for the interest in the insights it has provided on the structure of the human bulb. We have carefully considered the comments and responded to all points and revised the manuscript accordingly.

As a neuropathologist, I just have a couple of questions:

1. Post mortem human olfactory bulb tissue is notoriously fragile and the bulbs are often damaged on removal. The specimens used here appear to be intact but how did the investigators manage to maintain the tissue structural integrity through 10 cycles of their multiplex immunofluorescence involving multiple rinsing and coverslip removal steps?

The reviewer is correct that human olfactory bulbs are particularly fragile. Over many years our human brain bank in New Zealand has refined the process of collecting the olfactory bulb to minimise damage to the structure. From our previous work we have found that careful handling, formalin fixation and paraffin-embedding the bulbs preserves the structure well for immunohistochemistry studies (Zapiec et al., 2017, Murray et al., 2020, Highet et al., 2020, Stevenson et al., 2020). These factors and the use of charged slides ensured the tissue structure was preserved and bulb sections stayed adhered to the slide throughout the wash and coverslip removal steps. Care was taken to decoverslip by immersing the slide in PBS and allowing the coverslip to slide off without any force applied. Overall, we found the tissue to be very resilient to the iterative multiplex IHC labelling process.

2. While this is largely a technique paper I think it would be strengthened if the discussion on the findings regarding the novel anatomical observations and the anatomical specificity of the tau and α -synuclein pathology was expanded. There is a wealth of data in the supplementary material mapping the pathology in the different cases but a summary figure illustrating the differential distribution of the pathology in AD and PD cases would help the reader to see this. Can the authors speculate a little on why these differences occur?

As reviewer 1, 3 and 4 all requested additional discussion of the labelling we present in the context of previous literature, we have expanded our discussion of the points requested in lines 275-288, 337-344 and 349-373. In these additional discussion sections, we have commented on some of the potential reasons and implications of the changes we observed.

Reviewers' comments:

Reviewer #1 (Remarks to the Author):

The authors have answered each and every one of the questions. Thank you very much and congratulations for this study.

Reviewer #2 (Remarks to the Author):

In this work, multiple antibodies were used to stain target proteins, and their fluorescence signals were analyzed. Thus, the staining patterns of the antibodies used in the analysis should be strictly validated—however, Supplementary Fig 1, which shows the validation study of the antibody stripping technique, raises the concern. First, the same NeuN antibody was used in both round 2 and round 10, but no NeuN signals were observed in round 10. Do the imaging results depend on the staining round, even if the same antibody is used? A similar result was observed in MAP2. When comparing rounds 6 and 10, the MAP2 signal significantly decreased in round 10. Also, the fluorescence signal intensities of Collagen IV in rounds 1 and 9 showed a significant difference. Also, for SMA, the fluorescence signal intensities in rounds 1 and 9 are different. Tomato lectin was used in all rounds and showed a high-level variation in their fluorescence intensities. The antibody stripping technique can damage tissues. However, for some antibodies in Supplementary Fig. 1, the staining in round 1 or 2 showed lower fluorescence intensities than in round 9 or 10, which is not clear to this reviewer. If the fluorescence intensity depends on the staining round, more careful validation and analysis would be needed when the authors make a conclusion based on the results acquired using this staining technique. Can the authors guarantee that a similar level of fluorescence signals would be observed if the same antibody is used in the same round? This reviewer recommends performing a more detailed validation study on their antibodies and antibody stripping technique, at least for the antibodies used in their clinical study.

Reviewer #4 (Remarks to the Author):

The authors have addressed the points raised in my review and have added some appropriate text to expand discussion of their results.

Reviewer 2

In this work, multiple antibodies were used to stain target proteins, and their fluorescence signals were analyzed. Thus, the staining patterns of the antibodies used in the analysis should be strictly validated—however, Supplementary Fig 1, which shows the validation study of the antibody stripping technique, raises the concern. First, the same NeuN antibody was used in both round 2 and round 10, but no NeuN signals were observed in round 10. Do the imaging results depend on the staining round, even if the same antibody is used? A similar result was observed in MAP2. When comparing rounds 6 and 10, the MAP2 signal significantly decreased in round 10. Also, the fluorescence signal intensities of Collagen IV in rounds 1 and 9 showed a significant difference. Also, for SMA, the fluorescence signal intensities in rounds 1 and 9 are different. Tomato lectin was used in all rounds and showed a high-level variation in their fluorescence intensities. The antibody stripping technique can damage tissues. However, for some antibodies in Supplementary Fig. 1, the staining in round 1 or 2 showed lower fluorescence intensities than in round 9 or 10, which is not clear to this reviewer. If the fluorescence intensity depends on the staining round, more careful validation and analysis would be needed when the authors make a conclusion based on the results acquired using this staining technique. Can the authors guarantee that a similar level of fluorescence signals would be observed if the same antibody is used in the same round? This reviewer recommends performing a more detailed validation study on their antibodies and antibody stripping technique, at least for the antibodies used in their clinical study.

We believe that the combination of Supplementary Fig 1 (empirical validation of antibody panels) and Supplementary Fig 2 (summary of marker labelling included in the analysis) provide a comprehensive overview and validation of the antibodies used in the analysis. All antibodies used in this study have been previously validated and extensively referenced in the literature for use in formalin-fixed, paraffin-embedded rodent and human brain tissues, and also cited in our previous publications (Bogoslovsky et al., 2017; Highet et al., 2020; Maric et al., 2021; Murray et al., 2020; Stevenson et al., 2020; Swanson et al., 2020; Zapiec et al., 2017). We also provided the Research Resource Identifier (RRID) for each antibody (Supplementary Table 1) sourced from The Antibody Registry (<https://antibodyregistry.org/>), which provides registration information for each antibody used in this study. Furthermore, distribution of the labelling we observed for the antibodies used in the spatial analysis is consistent with previous reports and the single round labelling of these antibodies presented in our previous studies.

Regarding the difference in antibody labelling between different rounds of our multiplex IHC staining protocol, we are fully aware in changing fluorescence signal intensities when some of the antibodies were applied in different rounds, which we assessed to be dependent on the extent of antigen retrieval, antibody stripping and tissue damage over successive rounds. Therefore, we applied a systematic and entirely empirical approach, using incremental antigen retrieval conditions attained through iterative HIER steps in Tris-EDTA pH9 buffer, which were designed to progressively unmask target antigens in the same tissue samples, whereby the most heat-labile tissue targets could be probed in early rounds of multiplex IHC

staining, after minimal antigen retrieval steps, and more robust targets are probed in later rounds of multiplex IHC staining, after more extensive antigen retrieval steps. Given the rationale above, for some antibodies, the epitopes of interest were not optimally retrieved in the initial rounds, such as collagen IV and SMA in round 1, but were successfully retrieved with additional iterative rounds of HIER, as shown in round 9. For other antibodies, the epitope retrieval was optimal in earlier rounds and degraded in later rounds due to tissue damage after additional heating and antibody stripping steps, such is the case for MAP2 and NeuN in round 10.

We included some antibodies in multiple panels (NeuN, MAP2) as they were immunocompatible with the rest of the antibodies in the panel and helped interpret or empirically validate the labelling of other antibodies in those rounds prior to image alignment. As mentioned above, other antibodies were repeated across rounds if the signal in earlier rounds was not optimal (collagen IV, SMA) to see if further retrieval improved the signal. It is important to note that while signal to background intensity varied between the rounds, when labelling was successful for an antibody, the qualitative distribution of the labelling did not change and was comparable to that of single round labelling in our previous studies. For example, the pathology markers (tau AT8 and amyloid) were repeated in round 10 to confirm that additional HIER antigen retrieval had not unmasked any additional antigen as these antibodies are known to require extensive antigen retrieval.

For the antibodies that were repeated across several rounds and were included in the spatial analysis (histones, tomato lectin, MAP2, tau AT8), we selected the image that consistently showed the best signal compared to background across all cases. This was done by comparing the grayscale values for true signal and tissue background signal for each image. For every case, the image from the same round was used for analysis, i.e., MAP2 from round 6 was used for every case in the spatial analysis.

Regarding the quantification of labelling, our spatial analysis assessed the tissue area labelled by each antibody in the human bulb sections and did not quantitate the levels of these targets by measuring fluorescence signal intensities per pixel. Fluorescence signal intensity relative to background was used to threshold each marker for the pixel bin binary segmentation and this was internally normalized to the maximum intensity per label, per section. That is, the Poisson p-value was calculated per bin per label per section and bins having a p-value < 0.05 were coded "1" while those that did not were coded "0". We then compared the ratio of positive bins per label : total bins per section across the cases. This binary decision of epitope present in a pixel bin or not means that the quantitative value is not important as long as it was significantly above the background noise level.

In summary, any differences in fluorescence intensity of true signal between labelling rounds did not affect the comparison between cases because the image with optimal signal to background was selected, only images from the same round were compared across cases, the thresholding of the signal was internally normalised to the background of each section and the analysis compared the ratio of positive bins : total bins across each case rather than the fluorescence intensity of the labelling in those bins.

We acknowledge that there are limitations to this multiplexed labelling method regarding tissue damage and users should be aware that signal intensity will vary with the iterative labelling process. Therefore, we have added the following text to the discussion in line 384-399:

“Additional considerations are required for MP-IHC labelling in neuroanatomical studies. Delicate tissue may be susceptible to damage caused by the iterative processes of coverslip removal, antibody stripping, and antigen retrieval. Therefore, great care must be taken during these steps to preserve the tissue. Some antigens may also be degraded in the later rounds of labelling, leading to variable fluorescence intensity. Our incremental antigen retrieval conditions using iterative heat-mediated antigen retrieval steps in Tris-EDTA pH9 buffer were designed to progressively unmask target antigens in the same tissue samples, whereby the most heat-labile tissue targets could be probed in early rounds of MP-IHC, after minimal antigen retrieval steps, and more robust targets are probed in later rounds after more extensive antigen retrieval steps. It is therefore important to validate that the antibodies used produce the same labelling distribution in both single and multiple round labelling experiments. All antibodies used for this study have been previously validated for use in FFPE rodent and human brain tissues as cited in our previous publications^{8,11,12,18,27,50,51} and our MP-IHC labelling was consistent with that observed using single round labelling in these previous studies. For quantitative comparisons, tissue sections need to undergo the same labelling protocol and only labelling from the same round should be compared across different tissue sections.”